# Automatic Task-aware Instruction Optimizer for Black-box LLMs

## Abstract

Large Language Models (LLMs) have demonstrated superior capabilities in terms of solving various real-world tasks. However, their performance and generated content quality heavily depend on task-relevant instructions, which makes instruction optimization a challenging but critical direction to explore. In particular, as practitioners generally cannot access black-box (or API) LLMs' internal parameters and gradient information, it consequently makes instruction optimization for black-box LLMs especially non-trivial. Existing methods for optimizing black-box LLM instructions mainly focus on in-context learning using manually designed or heuristic disciplines, which can be insufficient due to the extreme complexity of modern black-box LLMs that can contain hundreds of billions of parameters. To address these challenges, we propose a novel automatic instruction optimization framework named Automatic Instruction Optimizer (AIO). AIO is designed to perceive target task information and adaptively adjust its task-aware instructing strategy for a task-solver black-box LLM. By leveraging a white-box LLM with parameter fine-tuning for enhanced representation power, AIO can automatically update its instructing strategy based on the feedback from task-solver black-box LLM. To achieve this goal, AIO adopts a novel LLM parameter fine-tuning process powered by zeroth-order gradient approximation and Contextual Bandit techniques, which can effectively and efficiently help address the challenge of inaccessible black-box LLM internal parameters and gradients, as well as help alleviate expensive API cost concerns by flexibly reusing collected black-box LLM feedback. Extensive empirical evaluations are presented to demonstrate properties of our proposed AIO, and its effectiveness in comparison with strong baselines.

## 1 Introduction

Large Language Models (LLMs) have demonstrated impressive performance across various application scenarios, such as knowledge graph reasoning (Pan et al., 2024). However, LLMs generally rely on well-crafted instructions that provide accurate guidance and sufficient reference for high-quality answer generation. Designing such instructions can be particularly challenging for more powerful black-box (or API) LLMs (e.g., GPT-4 (Achiam et al., 2023), Claude-3 (Anthropic, 2024)), as their parameters and gradients are commonly inaccessible. Meanwhile, effective instructing strategies can vary significantly across different LLMs, distinct LLM versions, and downstream tasks (Zhou et al., 2022; Khattab et al., 2023; 2022), while optimal instructing strategies generally require flexible adaptations tailored to target tasks or domains (Sun et al., 2024; Liu et al., 2024). In this case, designing an optimal instructing strategy for a specific target task like knowledge reasoning can be non-trivial and expensive. In addition, crafting proper instructing strategies for domain-specific tasks can be difficult and time-consuming for human experts (Brown et al., 2020; Reynolds & McDonell, 2021; Shin et al., 2020). For instance, assigning human labor to refine such task-specific instructions will be expensive, and the cost can grow exponentially along with increasingly more task categories of higher granularity levels (Scao & Rush, 2021; Shin et al., 2023; Amatriain, 2024). Thus, it is necessary and valuable to develop automatic task-aware instructing mechanisms based on task information to enable optimal performance of task-solver black-box LLMs, without intervention of human experts.

Regarding instruction optimization in terms of black-box LLMs, there is an emerging line of works using an additional instruction-generating LLM as an "instruction engineer" (Zhou et al., 2022; Pryzant et al., 2023; Fernando et al., 2023; Guo et al., 2024; Chen et al., 2024; Lin et al., 2024), in order to leverage the strong representation power of LLMs in search of a good instructing strategy.

However, existing works mainly focus on in-context learning aspects with frozen LLM parameters, based on manually crafted or heuristic disciplines, which can limit the ability of LLMs in terms of perceiving and utilizing target task information and black-box LLM feedback. On the other hand, LLM fine-tuning alternatively offers more elasticity by involving trainable LLM parameters to maintain an over-parameterization advantage (Allen-Zhu et al., 2019) with regard to exemplars from target tasks, which can allow LLMs to adapt their interpretations to align task textual data with target objectives (Han et al., 2024b) such as human preferences (Korbak et al., 2023; Rafailov et al., 2024), along with parameter-efficient fine-tuning options (Hu et al., 2021a; Wu et al., 2024; Han et al., 2024b). Meanwhile, as modern LLMs are commonly pre-trained to achieve good generalization abilities instead of being optimized for specific downstream tasks (i.e., task-aware instruction optimization in our case) out of the box, LLM parameter fine-tuning can generally offer more flexibility and better performance than in-context learning techniques for various downstream applications (Liu et al., 2022; Mosbach et al., 2023). However, as black-box LLM parameters and gradients are generally inaccessible, it is impractical to directly fine-tune the "instruction engineer" LLM with back-propagation based on chain rule and black-box LLM feedback.

In face of above motivations and challenges, we propose a novel framework named Automatic Instruction Optimizer (AIO) to automatically optimize instructions for task-solver black-box LLM, with regard to the target task. In particular, AIO composes and optimizes human-comprehensible instructions fed into a black-box LLM to improve its performance on the target task, for reinforced transparency and trustworthiness. To learn a good task-aware instructing strategy, distinct from existing in-context learning approaches, AIO alternatively fine-tunes a white-box LLM (e.g., Llama 3 (Dubey et al., 2024)) into a capable automatic instruction optimizer, which is able to perceive downstream task information and generate high-quality instructions accordingly. This formulation aims to tackle the formidable complexity of modern black-box LLMs (i.e., with possibly hundreds of billions of parameters involved). Here, with strong representation power and learning abilities of a fine-tuned white-box LLM, AIO is capable of learning the complex relationship between task-aware instructions and black-box LLM feedback. Intuitively, one significant obstacle is that black-box LLM parameters and gradients are commonly inaccessible. To address this challenge and achieve efficient gradient approximation for the black-box LLM, we propose a novel zeroth-order (ZO) gradient approximation approach aided by Thompson Sampling (TS), by modeling the ZO gradient approximation of the black-box LLM as a sequential decision-making process. In the meantime, during instruction optimization, it is necessary to retrieve black-box LLM feedback, which requires querying third-party APIs and incurs direct development costs. To alleviate API cost concerns, our TS-based ZO gradient approximation adaptively reuses collected black-box LLM feedback, enabling efficient instruction optimization through the rich representation power originated from white-box LLM fine-tuning. Our contributions can be summarized as:

- **Problem Formulation**: We delve into the realm of automatic task-aware instruction optimization for black-box LLMs, where existing in-context learning methods can fail to deliver optimal performance due to insufficient representation power of their learning models or mechanisms. Different from existing approaches, we alternatively transform the goal of instruction optimization into an LLM fine-tuning objective, where a white-box LLM with sufficient representation power is fine-tuned to generate high-quality task-aware instructions for a task-solver black-box LLM.

- **Proposed Framework**: Different from existing approaches where human experts are involved to manually design instructions for downstream tasks, our proposed AIO does not require such intervention of human experts, while finishing all the instruction optimization automatically through LLM fine-tuning. To enhance the trustworthiness of our instruction optimization process, AIO is able to optimize human-comprehensible instructions (i.e., instructions made up by concrete textual tokens) to provide additional insights and transparency for practitioners. To tackle challenges of inaccessible black-box LLM gradients and possibly expensive API costs of the black-box LLM, AIO utilizes a novel zeroth-order gradient approximation approach aided by Thompson Sampling. By inventively formulating ZO gradient approximation procedure as a sequential decision-making process, this design enables us to approximate black-box LLM gradients effectively and efficiently, which are essential for fine-tuning our instruction-generating white-box LLM.

- **Experiments**: Extensive experiments are conducted on real-world data sets, demonstrating the superior performance of AIO compared with state-of-the-art baselines, as well as efficiency advantages of AIO in terms of reducing API token costs. Furthermore, we perform additional analytical experiments to explore characteristics and properties of AIO, such as instruction optimization trajectory results that demonstrate how instructions evolve across the optimization process.

## 2 RELATED WORKS

**Instruction Optimization for API (Black-box) LLMs.** In contrast to white-box LLM instruction optimization (Shin et al., 2020; Li & Liang, 2021; Lester et al., 2021), as practitioners generally have no access to black-box LLM parameters or gradients, a line of existing works (Zhou et al., 2022; Prasad et al., 2022) perform instruction search based on manually defined criteria. Chen et al. (2024); Lin et al. (2024); Hu et al. (2024) also apply another LLM with frozen parameters to generate instructions for the black-box LLM, and gradually optimize the generated instruction based on Bayesian Optimization (Frazier, 2018; Wang et al., 2023; Shahriari et al., 2015), Contextual Bandit approaches (Chu et al., 2011; Li et al., 2010; Valko et al., 2013; Zhou et al., 2020; Agrawal & Goyal, 2013; Zhang et al., 2021), or localized instruction optimization guided by Gaussian Process (Schulz et al., 2018). Since these works primarily rely on manually designed or heuristic principles focused on in-context learning, they can result in sub-optimal black-box LLM performance. Alternatively, we fine-tune a white-box LLM into an "instruction engineer", capable of adaptively perceiving target task information and directly learning from black-box LLM feedback for instruction optimization.

**LLM-based Instruction Generation.** Instruction generation using LLMs is an emerging research topic (Zhou et al., 2022; Ma et al., 2024; Schnabel & Neville, 2024), where LLMs are applied as instruction optimizer and their instructing strategies are gradually refined based on environment feedback or target model outputs. In particular, there are a series of works leveraging meta-prompts, which can be manually designed by humans (Yang et al., 2024), or optimized by LLMs (Tang et al., 2024). Meanwhile, Pryzant et al. (2023) perform in-context "Gradient Descent" on instructions based on interactions with an instruction-generating LLM. Fernando et al. (2023); Guo et al. (2024) propose evolutionary algorithms to refine LLM-generated instructions in an in-context learning manner. Different from these works, our fine-tuned LLM can automatically perceive task-relevant information and black-box LLM feedback, which can generally offer more flexibility than in-context learning approaches and require no labor of human experts during the instruction optimization process.

## 3 PROBLEM FORMULATION

As mentioned in our Introduction, given a target task $\mathcal{T}$, two LLMs are involved in our pipeline: (1) black-box LLM $\mathcal{F}_B(\cdot)$ is applied for task-solving, i.e., generating answers for task queries. Here, the black-box LLM is considered as part of our learning objective, as we aim to learn optimized instructions to enable the black-box LLM to achieve optimal performance. (2) white-box LLM $\mathcal{F}_W(\cdot; \Theta_W)$ with trainable parameters $\Theta_W$ aims to generate and optimize human-comprehensible instructions, based on task $\mathcal{T}$ and feedback from $\mathcal{F}_B(\cdot)$. Suppose that target task $\mathcal{T}$ is associated with three data collections (i.e., query-label pairs $(X, Y)$) individually drawn from task $\mathcal{T}$: (1) training data $\mathcal{D}_{\text{Train}}$, which can also be named as task exemplars, will be fed into the white-box LLM as reference for generating high-quality task-specific instructions; (2) validation data $\mathcal{D}_{\text{Valid}}$ is applied for performance evaluation during the optimization; and (3) final evaluation will be conducted on a separate testing data set, which will remain unrevealed and inaccessible during the optimization stage for instruction-generating white-box LLM. Meanwhile, we denote $\mathcal{F}_W(\mathcal{D}_{\text{Train}}; \Theta_W)$ as the instruction generated based on exemplars $\mathcal{D}_{\text{Train}}$ and corresponding white-box LLM parameters $\Theta_W$. With generated instruction, denote $\hat{Y} = \mathcal{F}_B\big([\mathcal{F}_W(\mathcal{D}_{\text{Train}}; \Theta_W); X]\big)$ as the answer generated by black-box LLM for query $X$, where $[\cdot; \cdot]$ operation embeds query $X$ to the generated instruction.

**Learning Objective.** With task exemplars $\mathcal{D}_{\text{Train}}$ and an evaluation function (e.g., loss function) $\mathcal{L}(\cdot, \cdot)$, we transform the instruction optimization process into a white-box LLM fine-tuning objective, to leverage the sufficient learning and representation power of LLM fine-tuning for instruction optimization. Here, we aim to find the optimal white-box parameters $\Theta_W$ that minimize:

$$\min_{\Theta_W} \left[ \mathbb{E}_{(X,Y)\sim\mathcal{T}}\big[\mathcal{L}(\hat{Y}, Y)\big] \right] = \min_{\Theta_W} \left[ \mathbb{E}_{(X,Y)\sim\mathcal{T}}\big[\mathcal{L}\big(\mathcal{F}_B([\phi(\Theta_W); X], Y)\big)\big] \right] \quad (1)$$

in observation of task exemplars $\mathcal{D}_{\text{Train}}$, where we apply a shorthand for generated instruction:

$$\phi(\Theta_W) := \mathcal{F}_W(\mathcal{D}_{\text{Train}}; \Theta_W). \quad (2)$$

Intuitively, we utilize the above fine-tuning process to guide how white-box LLM comprehends task exemplars and composes task-specific instructions, based on task-solver black-box LLM feedback.

## 4 PROPOSED FRAMEWORK: AUTOMATIC INSTRUCTION OPTIMIZER (AIO)

Recall that we aim to fine-tune the white-box LLM parameters $\Theta_W$ to minimize our learning objective by optimizing generated instructions. An intuitive approach is to update the white-box LLM

Figure 1: Pipeline of AIO. The white-box LLM 🦙 generates an instruction ⟨⟩ from exemplars $\mathcal{D}_{\text{Train}}$, which is evaluated to produce validation loss. The gradient flow towards white-box LLM parameters is then decomposed into: (1) TS-aided ZO gradient approximation for black-box LLM gradients, and (2) back-propagation for white-box LLM gradients. Finally, white-box LLM parameters are updated via Gradient Descent.

parameters $\boldsymbol{\Theta}_W$ using gradients derived from the instruction evaluation results (Eq. 1). However, the nested black-box LLM makes direct back-propagation towards $\boldsymbol{\Theta}_W$ via the chain rule infeasible.

**Brief summary of AIO pipeline.** To address this challenge, we propose the AIO framework for efficient and effective white-box LLM fine-tuning, aimed at instruction optimization. As illustrated in Figure 1, the pipeline of AIO involves two major parts: (1) *Instruction Generation & Evaluation*: Given the exemplars $\mathcal{D}_{\text{Train}}$, the white-box LLM with parameters $\boldsymbol{\Theta}_W$ generates an instruction $\phi(\boldsymbol{\Theta}_W) := \mathcal{F}_W(\mathcal{D}_{\text{Train}}; \boldsymbol{\Theta}_W)$ as in Eq. 2. This instruction is then evaluated based on the black-box LLM output, which produces validation loss. (2) *White-box Instruction-generating LLM Fine-tuning*: Based on the validation loss, we can decompose the gradient flow towards $\boldsymbol{\Theta}_W$ into two multiplicative components via the chain rule: (i) white-box LLM gradients that can be obtained with back-propagation, and (ii) inaccessible black-box LLM gradients that are approximated with our proposed TS-aided ZO approximation method. Different from existing methods with randomly sampled ZO directions (e.g., Spall (1992); Malladi et al. (2023)), we apply TS here to adaptively determine the ZO directions for efficient and effective gradient approximation (Subsec. 4.1.2). We elaborate on technical details in Subsec. 4.1 and provide AIO pseudo-code in Algorithm 1.

**Validation loss.** Given the original optimization objective in Eq. 1, since a comprehensive overview of the task distribution $\mathcal{T}$ can be inaccessible, we alternatively evaluate the quality of generated instructions using validation data $\mathcal{D}_{\text{Valid}}$. This leads to our validation loss:

$$\mathcal{L}_{\text{Valid}}\big(\phi(\boldsymbol{\Theta}_W)\big) := \frac{1}{|\mathcal{D}_{\text{Valid}}|} \sum_{(X,Y)\in\mathcal{D}_{\text{Valid}}} \mathcal{L}\bigg(\mathcal{F}_B\big([\phi(\boldsymbol{\Theta}_W); X]\big), Y\bigg), \qquad (3)$$

where the validation loss is evaluated on the instruction $\phi(\boldsymbol{\Theta}_W)$, which is generated by the white-box LLM $\mathcal{F}_W(\cdot; \boldsymbol{\Theta}_W)$ as in Eq. 2. Consequently, the gradient flow towards white-box LLM parameters $\boldsymbol{\Theta}_W$ will become $\partial\left[\frac{1}{|\mathcal{D}_{\text{Valid}}|}\sum_{(X,Y)\in\mathcal{D}_{\text{Valid}}}\mathcal{L}\big(\mathcal{F}_B([\phi(\boldsymbol{\Theta}_W); X]), Y\big)\right]\big/\partial\boldsymbol{\Theta}_W$. Given the nested black-box LLM, we are unable to directly back-propagate towards $\boldsymbol{\Theta}_W$ through the chain rule.

## 4.1 THOMPSON SAMPLING (TS) AIDED ZEROTH-ORDER (ZO) GRADIENT APPROXIMATION

To deal with the challenge of inaccessible black-box LLM gradients, by applying chain rule on Eq. 3, we first can decompose the gradient flow with respect to white-box LLM parameters $\boldsymbol{\Theta}_W$ into two separate multiplicative parts: (1) gradients involving the black-box LLM; (2) and white-box LLM gradients that can be obtained by back-propagation, as

$$\partial\left[\frac{1}{|\mathcal{D}_{\text{Valid}}|}\sum_{(X,Y)\in\mathcal{D}_{\text{Valid}}}\mathcal{L}\big(\mathcal{F}_B([\phi(\boldsymbol{\Theta}_W); X]), Y\big)\right]\Big/\partial\boldsymbol{\Theta}_W$$

$$= \underbrace{\frac{\partial\left[\frac{1}{|\mathcal{D}_{\text{Valid}}|}\sum_{(X,Y)\in\mathcal{D}_{\text{Valid}}}\mathcal{L}\big(\mathcal{F}_B([\phi(\boldsymbol{\Theta}_W); X]), Y\big)\right]}{\partial\phi(\boldsymbol{\Theta}_W)}}_{\text{Black-box LLM Gradients}} \cdot \underbrace{\frac{\partial\phi(\boldsymbol{\Theta}_W)}{\partial\boldsymbol{\Theta}_W}}_{\text{White-box LLM Gradients}}. \qquad (4)$$

On one hand, while our white-box LLM gradients can be obtained by conventional back-propagation in a straightforward way, we can also readily integrate back-propagation with Parameter-efficient Fine-tuning (PEFT) techniques, such as LoRA (Hu et al., 2021a), to enhance the efficiency of white-box LLM fine-tuning while still maintaining promising performance. One the other hand, as we have mentioned, it is not plausible to directly derive the first term on the right-hand side with back-propagation, since black-box LLM $\mathcal{F}_B(\cdot)$ parameters and gradients are inaccessible. In this case, we propose to tackle this challenge with the zeroth-order gradient approximation technique.

### 4.1.1 ZEROTH-ORDER GRADIENT APPROXIMATION FOR BLACK-BOX LLM GRADIENTS

Zeroth-order gradient approximation has been proved effective and efficient for LLM fine-tuning (Malladi et al., 2023), yielding satisfactory results with only a few forward (i.e., inference) passes of LLMs. This makes zeroth-order gradient approximation a promising solution for approximating inaccessible black-box LLM gradients. To begin with, analogous to existing ZO approximation works (e.g., Nesterov & Spokoiny (2017); Ghadimi et al. (2016); Duchi et al. (2015); Shu et al. (2023); Malladi et al. (2023)), we can first suppose a *linear optimization landscape* around each white-box LLM output $\phi$. With notation from Eq. 3, it leads to $\mathcal{L}_{\text{Valid}}(\phi+z) \approx [\nabla_\phi \mathcal{L}_{\text{Valid}}(\phi)]^\intercal \cdot z + \mathcal{L}_{\text{Valid}}(\phi)$; and $z$ is a small perturbation applied to the predicted next-token distribution, for all tokens in the output $\phi$, specifically on *LLM-header output probabilities* (i.e., predicted distribution over the vocabulary). We also include supplementary explanations for auto-regressive generation and perturbation in Appendix C.1. Here, we slightly abuse the notation by using operations "$+$" and "$-$" to impose perturbation $z$ onto token-level output probabilities of each token from $\phi$. This formulation holds because gradients $\nabla_\phi \mathcal{L}_{\text{Valid}}(\phi) := \partial \mathcal{L}_{\text{Valid}}(\phi)/\partial\phi$ will stay uniform for all $\phi$ within the *linear* landscape.

Here, these small token-level perturbations are imposed to collect information of the optimization landscape, as small perturbations on token-level outputs can effectively change the auto-regressive generation process (Han et al., 2024a). Inspired by Malladi et al. (2023), we can formulate an approximation for black-box LLM gradients as

$$\frac{\partial\big[\mathcal{L}_{\text{Valid}}\big(\phi(\boldsymbol{\Theta}_W)\big)\big]}{\partial\phi(\boldsymbol{\Theta}_W)} \approx \frac{\big[\mathcal{L}_{\text{Valid}}\big(\phi(\boldsymbol{\Theta}_W)+\epsilon z\big) - \mathcal{L}_{\text{Valid}}\big(\phi(\boldsymbol{\Theta}_W)-\epsilon z\big)\big]}{2\epsilon} \cdot z \tag{5}$$

by deeming $\mathcal{L}_{\text{Valid}}(\cdot)$ as the validating evaluation function for generated instruction $\phi$ from Eq. 3. Here, $z \sim \mathcal{N}(0, I) \in \mathbb{R}^d$ stands for a random Gaussian perturbation vector, imposed on each token of the output $\phi$, thereby maintaining the same dimensionality as the token-level dimensionality of the output. $z$ also satisfies the isotropic condition $\mathbb{E}[zz^\intercal] = I_d$. Consequently, $d$ will correspond to the vocabulary size of the white-box LLM. The scaling parameter $\epsilon \in \mathbb{R}^+$ is used to control the perturbation intensity. This formulation intuitively follows the idea of bi-directional estimation of optimization landscape to perceive the optimization landscape from both directions (Spall, 1992; Malladi et al., 2023). In this way, the first term on the RHS of Eq. 4 can be approximated with only black-box LLM forward passes, without accessing its internal gradients or parameters.

**Remark 1.** *We only approximate black-box LLM gradients, instead of using ZO method to directly estimate whole gradient flow $\partial\big[\frac{1}{|\mathcal{D}_{\text{Valid}}|} \sum_{(X,Y) \in \mathcal{D}_{\text{Valid}}} \mathcal{L}\big(\mathcal{F}_B([\phi(\boldsymbol{\Theta}_W);X]), Y\big)\big]/\partial\boldsymbol{\Theta}_W$. The reason is that the error of zeroth-gradient method tends to grow along with the target dimensionality (Malladi et al., 2023). Since the size of white-box LLM parameters $\boldsymbol{\Theta}_W$ is generally much larger than white-box LLM output dimensionality, we choose to approximate $\partial\big[\mathcal{L}_{\text{Valid}}(\mathcal{F}_B(\phi(\boldsymbol{\Theta}_W)), Y)\big]/\partial\phi(\boldsymbol{\Theta}_W)$ instead for a lower approximation error. Related ablation study is presented in Appendix B.6.*

However, one potential drawback is that as perturbation vectors $z$ are randomly sampled (Eq. 5), gradient perturbation directions within the optimization landscape will be random and potentially inefficient (Cai et al., 2022). Thus, we propose to reuse collected feedback, by formulating above ZO-based fine-tuning process as an online sequential decision-making problem; and utilize Contextual Bandit techniques to effectively determine which perturbation directions are informative, beneficial and worth exploring, in terms of improving instruction quality and black-box LLM performance.

### 4.1.2 TS-AIDED SELECTION OF GRADIENT PERTURBATION DIRECTIONS

Recall that for ZO-based gradient approximation methods (e.g., Nesterov & Spokoiny (2017); Ghadimi et al. (2016); Duchi et al. (2015); Malladi et al. (2023)), it is common to suppose that we have a linear optimization landscape around the current optimization objective as in Eq. 5. In this case, as illustrated in Figure 2, with random Gaussian perturbation vector $z$, we have the radius of the supposed linear optimization landscape following a Chi-squared distribution with expected radius being $\mathbb{E}[\text{Radius}] = \mathbb{E}\big[\|\epsilon z\|_2\big] = \epsilon\sqrt{d}$.

**Leveraging the linear optimization landscape.** Within the supposed linear landscape, we can intuitively formulate this ZO optimization problem into a sequential decision-making process, where collected information can help choose perturbation directions $z$ in Eq. 5, rather than applying completely random $z$. As a natural solution, Contextual Bandit algorithms (Li et al., 2010; Agrawal & Goyal, 2013) are designed to identify the optimal choice among a set of candidate arms (i.e.,

actions) based on arm contextual information and historical records, while addressing the exploration-exploitation dilemma in sequential decision-making processes Auer et al. (2002). Under Contextual Bandit settings, and based on the gradient approximation formulation from Eq. 5, we define the arm reward $r \in \mathbb{R}$ for each perturbation direction $z \in \mathbb{R}^d$ (i.e., *candidate arm* in Contextual Bandit) as

$$r := \left[ \nabla_\phi \mathcal{L}_{\text{Valid}}(\phi) \right]^\mathsf{T} \cdot z \approx \frac{\mathcal{L}_{\text{Valid}}(\phi + \epsilon \cdot z) - \mathcal{L}_{\text{Valid}}(\phi - \epsilon \cdot z)}{2\epsilon}, \tag{6}$$

with respect to current white-box LLM output $\phi$. Intuitively, our formulation of arm reward echoes with our bi-directional ZO approximation formulated in Eq. 5. This formulation aims to quantify the benefit of descending towards perturbation direction $z$ (i.e., updating output $\phi$ towards direction $-z$).

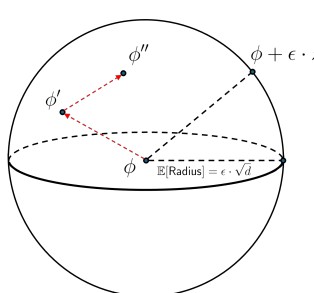

$$\mathcal{L}_{\text{Valid}}(\phi + \epsilon \cdot z) \approx [\nabla_\phi \mathcal{L}_{\text{Valid}}(\phi)]^\mathsf{T}(\epsilon \cdot z) + \mathcal{L}_{\text{Valid}}(\phi)$$

Figure 2: Linear optimization landscape (illustrated as a sphere) around white-box LLM output $\phi$. With the sampled perturbation direction $z \sim \mathcal{N}(0, \boldsymbol{I})$, expected radius will be $\epsilon\sqrt{d}$, in terms of $L_2$ distance between averaged output token-level probabilities. Updated outputs $\phi'$, $\phi''$ can stay within the linear landscape, motivating our TS-based approximation.

With our arm reward formulation enabled by the supposed linear optimization landscape, we apply a *linear* TS model to select perturbation directions (i.e., arms) accordingly. Here, TS model parameters with dimensionality $d$ are denoted by lowercase $\boldsymbol{\theta} \in \mathbb{R}^d$. Analogous to (Agrawal & Goyal, 2013; Zhang et al., 2021), we consider $\mathcal{N}(0, \boldsymbol{I})$ as the initial prior for TS model parameters. Then, starting from the prior, we gradually refine the corresponding TS parameter posterior distribution with collected optimization landscape information, and sample updated TS parameters $\boldsymbol{\theta}$ from the refined posterior.

Regarding our arm reward formulation in Eq. 6 and the nature of linear TS, the parameters $\boldsymbol{\theta}$ essentially serve as an estimate of the uniform gradients $\nabla_\phi \mathcal{L}_{\text{Valid}}(\phi)$ within the optimization landscape. In this way, we gradually refine our arm selection strategy by effectively reusing collected black-box LLM feedback. Notably, the linear optimization landscape allows us to employ a highly efficient linear TS algorithm for rapid arm selection and parameter refinement. Next, we proceed with arm (perturbation direction) selection for the ZO gradient approximation procedure.

**TS-aided perturbation direction (arm) selection.** We consider a $T$-round fine-tuning process, and denote the initial white-box LLM parameters without fine-tuning as $\boldsymbol{\Theta}_0 := \boldsymbol{\Theta}_W$, with initial generated instruction $\phi_0 := \phi(\boldsymbol{\Theta}_0)$. We also let $\boldsymbol{\Theta}_{t-1}, t \in [T]$ refer to white-box LLM parameters *before* $t$-th round fine-tuning. Here, in each round $t$, we first sample $K \in \mathbb{N}^+$ candidate arms (i.e., perturbation directions) $\mathcal{Z}_t$ for selection, from the standard Gaussian distribution, as

$$\mathcal{Z}_t := \left\{ z_{t,1}, z_{t,2}, \ldots, z_{t,K} \right\} \sim \mathcal{N}(0, \boldsymbol{I}). \tag{7}$$

This design controls the arm context norm magnitude with the isotropic formulation, while ensuring randomness in terms of candidate arm context directions. Before the fine-tuning process, initial TS model parameters are instantiated as $\boldsymbol{\theta}_0$ by sampling from the prior $\mathcal{N}(0, \boldsymbol{I})$. Here, we let $\boldsymbol{\theta}_{t-1}$ represent the TS model parameters in round $t$ before the refinement. We will discuss later how to update TS parameters using a refined TS parameter posterior in Eq. 9. Next, for these $K$ candidate arms, we formulate *estimated rewards* as the inner product $z_{t,k}^\mathsf{T} \boldsymbol{\theta}_{t-1}, \forall z_{t,k} \in \mathcal{Z}_t$, and select the top-$B$ arms with the *highest estimated rewards* to form collection $\widetilde{\mathcal{Z}}_t \subset \mathcal{Z}_t$, with cardinality $|\widetilde{\mathcal{Z}}_t| = B, B \ll K$. The chosen arms $\widetilde{\mathcal{Z}}_t$ are considered perturbation directions that can lead to potential benefits for reducing the validation loss. Calculation details are in lines 5-7 of Algorithm 1.

**Querying arm rewards.** Next, we query the black-box LLM (i.e., reward oracle) for validation loss results $\mathcal{L}_{\text{Valid}}$ to obtain rewards for the chosen arms $\widetilde{\mathcal{Z}}_t$. Using the shorthand $\phi_{t-1} := \phi(\boldsymbol{\Theta}_{t-1})$, for each chosen arm $z_{t,k} \in \widetilde{\mathcal{Z}}_t$, we calculate its arm reward by following Eq. 6, leading to

$$r_{t,k} = \frac{\mathcal{L}_{\text{Valid}}(\phi_{t-1} + \epsilon \cdot z_{t,k}) - \mathcal{L}_{\text{Valid}}(\phi_{t-1} - \epsilon \cdot z_{t,k})}{2\epsilon}, \tag{8}$$

which measures the benefit of involving direction (arm) $z_{t,k}$ for gradient approximation. Naturally, with each chosen arm (perturbation direction), the queried validation loss results are recycled to derive the black-box LLM gradient approximation according to Eq. 5. Finally, by plugging in the estimated black-box LLM gradients, *white-box LLM parameters can be updated via Gradient Descent through the gradient flow (Eq. 4) and estimated black-box LLM gradients (Eq. 5)*. Gradient estimation results from the $B$ chosen perturbation direction vectors (arms) are averaged for a more accurate approximation, analogous to existing ZO approximation methods (Malladi et al., 2023).

**Validating optimization landscape.** After querying arm rewards with Eq. 8 and performing white-box LLM parameter fine-tuning, we proceed to update the TS model parameters. Recall that we operate within a linear optimization landscape, if generated instructions do not significantly deviate from previous ones. In this case, we apply a threshold parameter $\beta > 0$ to quantify the landscape magnitude. For the optimized instruction $\phi_t$ in round $t$, we use the condition $\|\phi_t - \phi_{\text{Check}}\| > \beta$ to verify whether the assumed linear landscape remains valid. The initial checkpoint $\phi_{\text{Check}}$ is set as the instruction $\phi_0$ prior to optimization. We calculate the $L_2$ distance between the averaged token-level probabilities of the output $\phi_t$ and the checkpoint $\phi_{\text{Check}}$, inspired by prior works (e.g., Joshi et al. (2023); Manakul et al. (2023)). Collected records are initialized as an empty set $\Omega_0$.

**Updating TS parameters.** In each optimization round $t \in [T]$, we have *Scenario (1)*: If white-box LLM output becomes far enough from the checkpoint, s.t. $\|\phi_t - \phi_{\text{Check}}\| > \beta$, our current knowledge can be invalid because current white-box LLM output has significantly deviated from the checkpoint. In this case, we set the new checkpoint as $\phi_t$ and discard collected records. Then, reinitialize TS parameters $\boldsymbol{\theta}_t$ from the prior $\mathcal{N}(0, \boldsymbol{I})$. *Scenario (2)*: Otherwise, if the distance is small enough s.t., $\|\phi_t - \phi_{\text{Check}}\| \leq \beta$, the chosen arms and their true rewards from this round will be integrated into collected records $\Omega_t$. Afterwards, analogous to existing TS methods (Agrawal & Goyal, 2013; Zhang et al., 2021), with an exploration parameter $\nu \geq 0$, covariance matrix $\boldsymbol{\Sigma}_t := \boldsymbol{I} + \sum_{(\boldsymbol{z},r)\in\Omega_t} \boldsymbol{z} \cdot \boldsymbol{z}^{\mathsf{T}}$, and reward vector $\boldsymbol{b}_t := \sum_{(\boldsymbol{z},r)\in\Omega_t} \boldsymbol{z} \cdot r$, we update the posterior of TS parameters as

$$\mathcal{N}(\boldsymbol{\Sigma}_t^{-1}\boldsymbol{b}_t, \ \nu \cdot \boldsymbol{\Sigma}_t^{-1}). \tag{9}$$

Finally, we update TS parameters $\boldsymbol{\theta}_t$ by sampling from the updated posterior, and proceed to the next optimization round. Additional calculation details are presented in lines 12-19 of Algorithm 1.

**Remark 2.** *To reduce computational costs of matrix inversion and sampling from high-dimensional Gaussian distribution, motivated by Johnson-Lindenstrauss (JL) Lemma (Johnson & Lindenstrauss, 1984), we adopt random Gaussian projection to map $d$-dimensional arm contexts into a lower-dimensional space (Matoušek, 2008; Larsen & Nelson, 2017), where we perform the TS-aided selection of candidate arms $\mathcal{Z}_t$. Comparable ideas are also applied in existing works for reducing the dimensionality of tunable soft prompt vectors (Chen et al., 2024; Lin et al., 2024). To efficiently compute the inversion of the covariance matrix $\boldsymbol{\Sigma}_t$ in each round $t$, we utilize the Sherman-Morrison formula (Bartlett, 1951; Maponi, 2007), avoiding direct matrix inversion operations. Details will be elaborated in Appendix C.2 due to page limit.*

## 4.2 WORKFLOW SUMMARY AND PSEUDO-CODE FOR AIO FRAMEWORK

The pseudo-code of AIO is in Algorithm 1. For each optimization round $t \in [T]$, we first sample a pool of $K$ candidate arms (gradient approximation directions) $\mathcal{Z}_t$ (line 5, Algorithm 1). Then, we apply a TS-based bandit model to estimate arm rewards, which quantify the benefit of including the corresponding arms as perturbation directions. To reduce API costs, we only select $B \ll K$ arms as the chosen arms $\widetilde{\mathcal{Z}}_t \subset \mathcal{Z}_t$ (lines 6-7). Next, we query the black-box LLM to obtain rewards of the chosen arms (line 9) and perform Gradient Descent to fine-tune the white-box LLM parameters with the gradient flow described in Eqs. 4 and 5 (line 10). Afterwards, we check if the white-box LLM output after fine-tuning differs sufficiently from the checkpoint. If so, we reset the records and the TS parameter distribution (line 13). Otherwise, we update the parameter posterior with the chosen arms and their true rewards (lines 16-17). TS parameters $\boldsymbol{\theta}_t$ are updated accordingly (lines 14, 18).

## 5 EXPERIMENTS

Our empirical analysis mainly aims to show that AIO can effectively optimize task-specific black-box LLM instructions compared with strong baselines, as well as provide insights on AIO behaviours and properties. In terms of LLM implementations, we apply `Llama-3-8B-Instruct` (Dubey et al., 2024) as our tunable white-box LLM $\mathcal{F}_W(\cdot; \boldsymbol{\Theta}_W)$, and adopt `Claude-3-Sonnet` (Anthropic, 2024) as our black-box LLM $\mathcal{F}_B(\cdot)$. As an outline for our empirical results in the main body, we have: (1) zero-shot instruction induction experiments on 15 tasks in Subsec. 5.1; (2) empirical analysis on API token costs and performance in Subsec. 5.2; (3) a case study that provides analysis and examples for AIO instruction optimization trajectories in Subsec. 5.3. Due to page limit, we include experiment and implementation details in Appendix A. Meanwhile, we also conduct additional experiments presented in Appendix B, including but not limited to few-shot instruction induction results, ablation study on AIO, applying AIO under Chain-of-Thought (CoT) settings, as well as empirical results with different combinations of white-box LLMs and black-box LLMs.

---

**Algorithm 1** Automatic Instruction Optimizer (AIO)

---

1: **Input:** Optimization rounds $T$. Exemplars $\mathcal{D}_{\text{Train}}$, validation data $\mathcal{D}_{\text{Valid}}$. Number of candidate arms $K$. Number of chosen arms $B$. Exploration parameter $\nu \geq 0$. Threshold parameter $\beta > 0$.

2: **Initialization:** TS Model Parameters $\boldsymbol{\theta}_0 \sim \mathcal{N}(0, \boldsymbol{I})$. White-box LLM parameters $\boldsymbol{\Theta}_0 \leftarrow \boldsymbol{\Theta}_W$. Instruction checkpoint $\phi_{\text{Check}} \leftarrow \phi(\boldsymbol{\Theta}_0)$. TS model records $\Omega_0 \leftarrow \emptyset$.

3: **for** each round $t \in [T]$ **do**

4:     $\triangleright$ - - - - - - - - - - - - **TS-aided ZO Perturbation Direction Selection** - - - - - - - - - - - -

5:     Sample candidate perturbation directions (i.e., candidate arms) $\mathcal{Z}_t$ (Eq. 7), with $|\mathcal{Z}_t| = K$.

6:     Calculate arm reward estimations $\hat{r}_{t,k} = \boldsymbol{z}_{t,k}^{\mathsf{T}} \boldsymbol{\theta}_{t-1}, \forall \boldsymbol{z}_{t,k} \in \mathcal{Z}_t$, with TS parameters $\boldsymbol{\theta}_{t-1}$.

7:     Choose $B$ arms of highest estimated rewards $\widetilde{\mathcal{Z}}_t \leftarrow \arg\max_{\widetilde{\mathcal{Z}}_t \subset \mathcal{Z}_t : |\widetilde{\mathcal{Z}}_t| = B} \left[ \sum_{\boldsymbol{z}_{t,k} \in \widetilde{\mathcal{Z}}_t} \hat{r}_{t,k} \right]$.

8:     $\triangleright$ - - - - - - - - - - - - - **White-box LLM Parameter Fine-tuning** - - - - - - - - - - - - - -

9:     Query rewards for chosen arms $\widetilde{\mathcal{Z}}_t$ (Eq. 8).    $\triangleright$ **Only query chosen arms to reduce API cost.**

10:     With $B$ chosen perturbation directions (arms) $\widetilde{\mathcal{Z}}_t$ and their queried rewards, fine-tune white-box LLM parameters to $\boldsymbol{\Theta}_t$ with Gradient Descent, based on gradient flow decomposition (Eq. 4 and Eq. 5) and validation loss (Eq. 3). Generate updated instruction $\phi_t := \phi(\boldsymbol{\Theta}_t)$.

11:     $\triangleright$ - - - - - - - - - - - - - - - - - - **Updating Linear TS Model** - - - - - - - - - - - - - - - - - -

12:     **if** $\|\phi_t - \phi_{\text{Check}}\| > \beta$ **then**

13:        Reset prior as $\mathcal{N}(0, \boldsymbol{I})$, new checkpoint $\phi_{\text{Check}} \leftarrow \phi_t$, and collected records $\Omega_t \leftarrow \emptyset$.

14:        Sample updated TS parameters $\boldsymbol{\theta}_t \sim \mathcal{N}(0, \boldsymbol{I})$.

15:     **else**

16:        With chosen arms $\boldsymbol{z} \in \widetilde{\mathcal{Z}}_t$ and their rewards $r$, update $\Omega_t \leftarrow \Omega_{t-1} \cup \left[ \bigcup_{\boldsymbol{z} \in \widetilde{\mathcal{Z}}_t} (\boldsymbol{z}, r) \right]$.

17:        Update the posterior for TS parameters by $\mathcal{N}(\boldsymbol{\Sigma}_t^{-1} \boldsymbol{b}_t, \nu \boldsymbol{\Sigma}_t^{-1})$ (Eq. 9).

18:        Sample updated TS parameters from the posterior distribution $\boldsymbol{\theta}_t \sim \mathcal{N}(\boldsymbol{\Sigma}_t^{-1} \boldsymbol{b}_t, \nu \boldsymbol{\Sigma}_t^{-1})$.

19:     **end if**

20: **end for**

---

## 5.1 EXPERIMENTS ON ZERO-SHOT INSTRUCTION INDUCTION

We first experiment on zero-shot instruction induction performance of AIO. Analogous to existing works for LLM instruction optimization (Zhou et al., 2022; Chen et al., 2024; Lin et al., 2024), our empirical analysis involves 15 different tasks including instruction induction tasks from Honovich et al. (2022), as well as more challenging reasoning tasks from BigBench (bench authors, 2023). Consequently, evaluation criteria will vary depending on specific tasks, such as "Multiple Choice" where the white-box LLM needs to choose the right option out of several candidates, and "Exact Match" where black-box LLM answers needs to be identical to ground-truth labels. We defer detailed task descriptions and evaluation criteria to Appendix A.1.

**Baseline methods.** We involve four baselines, including two kinds of LLM-based instruction optimization methods. The first two baselines leverage black-box LLM for instruction generation: (1) APE (Zhou et al., 2022), (2) ProTeGi (Pryzant et al., 2023). We also include baselines that utilize white-box LLM for instruction generation: (3) InstructZero (Chen et al., 2024), (4) INSTINCT (Lin et al., 2024). Detailed baseline descriptions are included in Appendix B. Here, we apply `Claude-3-Sonnet` as the black-box instruction generation LLM for APE and ProTeGi, while adopting white-box LLM `Llama-3-8B-Instruct` for InstructZero and INSTINCT.

**Two PEFT variants of AIO.** Apart from the original AIO framework in Algorithm 1, recall that AIO is also compatible with many existing PEFT methods for efficient white-box LLM fine-tuning. Therefore, to reinforce fine-tuning efficiency, we further include empirical results of incorporating Linear Probing (LP) (Kumar et al., 2022) and LoRA (Hu et al., 2021a) to our proposed AIO. These two variants are denoted as "AIO + LP" and "AIO + LoRA" respectively. In particular, we note that "AIO + LP" only fine-tunes $\sim 6.54\%$ of white-box LLM parameters, while "AIO + LoRA" merely needs to fine-tune $\sim 0.04\%$ of white-box LLM parameters.

**Empirical results.** Our empirical results are shown in Table 1. We notice that our proposed AIO can generally achieve better performance in comparison with strong baselines, in the presence of the challenging reasoning tasks from BigBench (bench authors, 2023). In particular, our light-

| Tasks \ Methods | Black-box LLM | | White-box LLM | | White-box LLM w/ FT (Ours) | | |
|---|---|---|---|---|---|---|---|
| | APE | ProTeGi | InstructZero | INSTINCT | AIO | AIO + LP | AIO + LoRA |
| antonyms | 0.893 | 0.861 | 0.843 | 0.881 | **0.901** | 0.857 | 0.898 |
| sentiment | 0.911 | 0.928 | 0.941 | 0.920 | 0.949 | **0.967** | 0.947 |
| larger_animal | 0.914 | 0.932 | 0.827 | 0.857 | 0.912 | 0.945 | **0.950** |
| taxonomy_animal | 0.491 | 0.970 | 0.598 | 0.782 | **0.983** | 0.979 | 0.935 |
| object_counting | 0.319 | **0.550** | 0.522 | 0.537 | 0.543 | 0.401 | 0.479 |
| navigate | 0.580 | 0.624 | 0.556 | 0.577 | **0.644** | 0.623 | 0.627 |
| winowhy | 0.022 | 0.703 | 0.671 | **0.725** | 0.622 | 0.646 | 0.635 |
| implicatures | 0.806 | 0.826 | 0.816 | 0.837 | 0.811 | 0.836 | **0.849** |
| logical_fallacy | 0.820 | 0.826 | 0.790 | 0.826 | **0.868** | 0.824 | 0.836 |
| hyperbaton | 0.515 | 0.499 | 0.467 | 0.502 | **0.538** | 0.518 | 0.527 |
| epistemic_reasoning | 0.604 | 0.459 | 0.667 | 0.580 | 0.766 | **0.784** | 0.719 |
| movie_recommendation | 0.348 | 0.847 | 0.895 | 0.866 | **0.902** | 0.857 | 0.883 |
| timedial | 0.532 | 0.718 | 0.786 | 0.712 | **0.814** | 0.734 | 0.759 |
| presuppositions_as_nli | 0.458 | 0.488 | 0.503 | 0.482 | **0.523** | 0.486 | 0.493 |
| question_selection | 0.712 | 0.667 | **0.718** | 0.605 | 0.648 | 0.628 | 0.622 |
| Average Rank | 3.87 | 2.80 | 3.13 | 3.20 | **2.00** | | |

Table 1: Zero-shot Instruction Induction Results. For each task (row), **bold** number refers to the best result, while underlined number refers to the second-best one. AIO and its two variants can outperform four baselines on 12 out of a total of 15 tasks. We average results of AIO and its two variants task-wise, and treat these three methods as a unified baseline for ranking comparisons.

weight variants "AIO + LP" and "AIO + LoRA", which fine-tune significantly less white-box LLM parameters, can also achieve promising performance, or outperform other baselines on tasks like "epistemic reasoning". To perform a more comprehensive ranking comparison, we present ranking results by averaging the performance of AIO and its two variants for each task, and applying averaged performance for comparing against baselines to derive ranking. As in Table 1, AIO and its variants as a unity can still enjoy relatively higher ranking results compared with baselines. We also include additional complementary experiments, such as few-shot instruction induction and an ablation study of AIO components, so that interested readers can refer to Appendix B for details.

## 5.2 OPTIMIZATION PERFORMANCE VERSUS API TOKEN EFFICIENCY

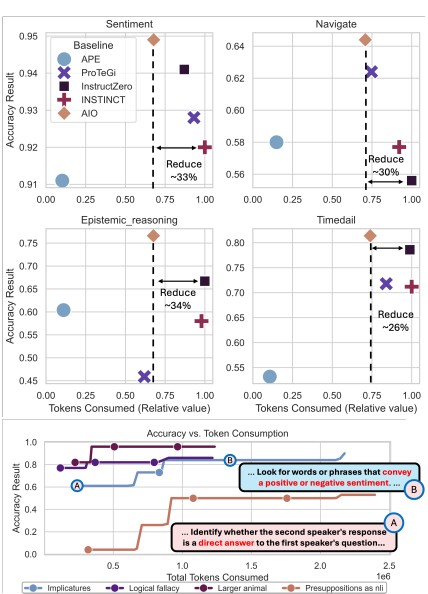

Figure 3: Token consumption vs. performance (accuracy results). Token consumption results are normalized into [0, 1] range. In the below figure, we include token consumption vs. best accuracy results till certain token consumption levels on four tasks, with two instruction fragments for "Implicatures" task at different stages.

Recall that we apply `Claude-3-Sonnet` as our black-box LLM $\mathcal{F}_B(\cdot)$ for experiments, where API query costs are charged on a token-basis for end users. In Figure 3, we show an illustration in terms of token consumption, with input and output token quantities combined, versus instruction induction performance on four different tasks. From Figure 3, we see that AIO can relatively maintain a good balance between token costs and induction performance, starting from early optimization stages when small amounts of tokens are consumed. Compared with ProTeGi, the performance of APE tends to be inferior as ProTeGi can optimize generated instructions with higher-granularity error feedback in terms of specific training samples, although it can lead to additional token consumption costs. On the other hand, we observe that the baselines InstructZero and INSTINCT generally have higher token consumption compared to AIO. While these two baselines also leverage white-box LLMs for instruction optimization, their methods primarily rely on in-context learning, by tuning a prefix soft prompt with kernel-based learner or small neural model. Given the extreme complexity of black-box LLMs, their representation power can be insufficient for effectively learning from black-box LLM feedback, leading to more interactions with the black-box LLM and, consequently, higher token costs. Alternatively,

Figure 4: In left Figure, we show three instructions generated for "Sentiment" task: w/o FT, during FT, and after FT. Instructions with FT generalize to task contexts instead of over-fitting task exemplars. Middle and right figures shows the normalized distance vs. accuracy for tasks "Larger Animal", "Question Selection" (middle figure with fitted lines), "Navigate" (right radar plot). For radar plot, instruction (i.e., point) accuracy increases clock-wisely starting from $0°$. Point distances to circle center are corresponding distances from $\phi_0$.

to tackle the complexity of the black-box LLM, we fine-tune a white-box LLM to leverage its rich representation power, enabling efficient instruction optimization with adequate utilization of black-box LLM feedback.

## 5.3 CASE STUDY: INSTRUCTION OPTIMIZATION TRAJECTORY ANALYSIS

In this subsection, we present analysis on how AIO generated instructions evolve across optimization rounds. From the left figure in Figure 4, we provide some examples in terms of how fine-tuning changes the way our white-box LLM $\mathcal{F}_W$ comprehends exemplars and composes instructions. These examples all originate from our experiment data. Without fine-tuning, $\mathcal{F}_W$ can over-fit exemplars by searching specific keywords to judge sentiment outcomes, which can clearly fail to generalize to unseen task data points and lead to low accuracy. During and after fine-tuning, $\mathcal{F}_W$ perceives black-box LLM feedback and corrects its way in terms of interpreting task exemplars as well as generating instructions. During fine-tuning, instruction gets improved by considering signal words from exemplars (e.g., "entertainingly") as *examples* instead of *sole indicators*. Moreover, after fine-tuning, $\mathcal{F}_W$ further generalizes the concept "positive connotations = positive sentiment" to a more comprehensive view, by mentioning "use sentiment analysis" to analyze outcome directly.

In addition, we also present visualization results in terms of induction performance as well as distances between generated instructions $\phi$ and the initial instruction $\phi_0 := \phi(\Theta_W)$. Instruction distances are measured by averaged token probabilities as in Algorithm 1. With the middle line chart, we individually plot a fitting line for instruction points of each task. For tasks like "Question Selection", our optimization trajectory tends to fit well with the supposed linear optimization landscape where small residuals are observed. On the other hand, for tasks like "Larger Animal", we can observe relatively higher fluctuations when generated instructions deviate from $\phi_0$. In this case, as it still obeys a relatively fitting linear trend until the normalized distance reaches $0.7 \sim 0.8$, we can supposed a smaller linear landscape in terms of TS-aided optimization. Shown in the radar plot, regarding the "Navigate" task, we see that good instructions are not necessarily far away from the initial $\phi_0$, while the best instructions tend to fall into a sub-area within the optimization landscape, instead of consistently deviates from the starting point $\phi_0$ as accuracy results increase. Due to page limit, we will include additional instruction optimization trajectory examples in Appendix B.12.

## 6 CONCLUSION

In this paper, we propose a novel framework named Automatic Instruction Optimizer (AIO) to adaptively customize instructions for various downstream tasks. By applying a task-solver black-box LLM for query answering, AIO fine-tunes a white-box LLM into a task-aware instruction optimizer that learns from high-level task-relevant information and black-box LLM feedback, to generate high-quality instructions for the task-solver black-box LLM. Distinct from existing in-context learning approaches, our framework is designed to address the formidable complexity of modern black-box LLMs with possibly hundreds of billions of parameters involved. To overcome the challenges of inaccessible black-box LLM gradients and mitigate concerns related to expensive black-box LLM API costs, our AIO framework leverages a novel TS-aided zeroth-order gradient approximation method, enabling effective and efficient learning of task-aware instructing strategies. Extensive experiments demonstrate the superiority of our proposed framework in terms of performance and API token efficiency, along with additional analyses that highlight AIO's properties and specifications. Additional discussions on AIO future extensions are presented in Appendix E.

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

# A  Complementary Details for Implementation and Experiments

## A.1  Descriptions for tasks involved in our experiments

| Task | Metric | Descriptions |
|---|---|---|
| antonyms | Exact Match | Find the antonym for the given word. |
| sentiment | Binary Choice | Judge the sentiment preference of the given review. |
| larger_animal | Binary Choice | Identify which of the input animals is larger. |
| taxonomy_animal | Exact Set | Identify all the animal words out of input word sequence |
| object_counting | Exact Match | Enumerate objects of different types and output the total number. |
| navigate | Binary Choice | Given a series of navigation instructions, determine whether one would end up back at the starting point. |
| winowhy | Binary Choice | Evaluate the reasoning in answering Winograd Schema Challenge questions. |
| implicatures | Binary Choice | Predict whether Speaker 2's answer to Speaker 1 counts as a yes or as a no. |
| logical_fallacy | Binary Choice | Detect informal and formal logical fallacies. |
| hyperbaton | Binary Choice | Order adjectives correctly in English sentences. |
| epistemic_reasoning | Binary Choice | Determine whether one sentence entails the next. |
| movie_recommendation | Multiple Choice | Recommend a movie that is similar to the given list of movies. |
| timedial | Multiple Choice | Pick the correct choice for a masked (temporal) span given the dialog context. |
| presuppositions_as_nli | Multiple Choice | Determine whether the first sentence entails or contradicts the second. |
| question_selection | Multiple Choice | Given a short answer along with its context, select the most appropriate question which has the given short answer as its answer. |

Table 2: Task descriptions and corresponding metrics.

For evaluation metrics: we have (1) Exact Match: the generated answer needs to exactly match the label; (2) Multiple Choice: task-solver LLM needs to choose one correct option out of several given candidate choices; (3) Binary Choice: task-solver LLM needs to choose one correct option out of two candidate choices; (4) Exact Set: whether the predicted set of items (e.g., animals) exactly matches the label set in both content and size, regardless of the item order.

## A.2  Templates applied for AIO instruction generation and black-box LLM inference (zero-shot induction and few-shot induction)

For zero-shot instruction induction and few-shot instruction induction, after obtaining generated instructions, we follow analogous ideas as in Zhou et al. (2022); Chen et al. (2024); Lin et al. (2024) when designing the instruction induction templates.

- Few-shot instruction induction settings:

```
<examples>
Exemplary data: [Exemplar data ([DEMO_DATA])]
</examples>.
Instruction: [INSTRUCTION]\n\n
Input: [Query INPUT]\n Output: [OUTPUT Placeholder]
```

- Zero-shot instruction induction settings:

```
 Instruction: [INSTRUCTION]\n\n
Input: [Query INPUT]\n Output: [OUTPUT Placeholder]
```

Obviously, the main difference between these two templates is that few-shot template will also involve task exemplars during the inference stage, which can provide additional reference for the task-solver black-box LLM.

# B  Complementary Experiments

Due to strict page limit for the main body, we choose to include complementary experiments here in this section. As an outline, we have (1) few-shot experiments on our 15 tasks located in Subsec. B.3;

(2) Chain-of-Thought (CoT) experiments in Subsec. B.4; (3) experiments with different combinations of white-box and black-box LLMs in Subsec. B.5; (4) an ablation study on AIO components in Subsec. B.6; (5) Parameter study for $\beta$ and $B$ in Subsec. B.7; (6) Empirical results with additional kinds of white-box LLMs in in Subsec. B.8; (7) Effects of exemplar quantity in Subsec. B.9; (8) Transferability across black-box LLMs in Subsec. B.10; (9) Empirical comparison with an additional baseline EvoPrompt in Subsec. B.11; (10) Additional optimization trajectory results in Subsec. B.12.

## B.1 BASELINE DESCRIPTIONS.

Recall that we involve four baselines for comparison, including two kinds of methods that utilize LLMs for instruction optimization. The first kind methods leverages black-box LLM for instruction generation: (1) APE (Zhou et al., 2022) which generates instruction using another *black-box* LLM with designated templates and instruction search mechanism; (2) ProTeGi (Pryzant et al., 2023) applies a black-box LLM for instruction generation, and optimizes "Gradient Descent" on the generated instruction by integrating with another *black-box* LLM. Meanwhile, we also include (ii) Methods that utilize white-box LLM for instruction generation: (3) InstructZero (Chen et al., 2024) generates instructions using a *white-box* LLM, while controlling the generation process by optimizing a prefix soft prompt, based on a kernel-based Bayesian Optimization approach; (4) INSTINCT (Lin et al., 2024) adopts an analogous approach as InstructZero, but alternatively applies a neural bandit model in replace of the kernel-based Bayesian Optimization for soft prompt selection.

## B.2 IMPLEMENTATION DETAILS

For our zero-shot and few-shot instruction induction experiments, we consider each task is associated with 20 task exemplars denoted as $\mathcal{D}_{\text{Train}}$ as well as 100 validation samples $\mathcal{D}_{\text{Valid}}$, which will remain the same for AIO and all other baseline methods. For AIO, when choosing our threshold parameter $\beta$, we initially set $\beta$ as an infinitely large value to enable all collected black-box LLM feedback to be reused. Afterwards, we will experiment with $\beta = \mathcal{O}(\epsilon\sqrt{d})$ and choose the constant for $\mathcal{O}(\cdot)$ notation with grid search $\{1, 10, 100\}$. We perform the fine-tuning process for $T = 10$ rounds, as well as set the exploration parameter $\nu = 0.1$ for all experiments For the perturbation magnitude parameter $\epsilon$, we choose its value with grid search from $\{10^{-3}, 10^{-4}, 10^{-5}\}$. In each optimization round, we will draw $K = 10000$ arms from $\mathcal{N}(0, \boldsymbol{I})$ where we choose $B = 3$ arms for fine-tuning white-box LLM as well as update the TS model parameters. Regarding our TS model, after applying JL-Lemma and random Gaussian matrix projection (Matoušek, 2008; Larsen & Nelson, 2017) for dimension reduction, we will have the reduced dimension of TS model to be approximately $d' \approx 10^4$, which leads to $\sim 0.4$ seconds for selecting chosen arms $\widetilde{\mathcal{Z}}_t$ and $\sim 3$ seconds for TS model parameters update in each round $t$. For "AIO + LoRA", we set its "intrinsic rank" of low-rank approximation to 8. As we mentioned in the main body, we apply `Llama-3-8B-Instruct` (Dubey et al., 2024) as our tunable white-box LLM $\mathcal{F}_W(\cdot; \boldsymbol{\Theta}_W)$, and adopt `Claude-3-Sonnet` (Anthropic, 2024) as our black-box LLM $\mathcal{F}_B(\cdot)$. All experiments are performed on a server with Intel Xeon CPU and NVIDIA V100 GPUs.

## B.3 COMPLEMENTARY EXPERIMENTS WITH FEW-SHOT AIO

In this subsection, we include experiment results that examine AIO performance under few-shot settings, where training samples (or exemplars) will be provided to black-box LLM for reference. In this case, generated instructions will need to provide high-level reference and guidance, to assist the answer generation of task-solver black-box LLM in observation of task exemplars. The template applied for experiment is also shown in Subsec. A.2.

The experiment results are shown in Table 3. Here, we see that under few-shot settings, there exist performance improvements for most baselines compared with zero-shot settings, due to the help of additionally available task exemplars. By leveraging the sufficient representation power of fine-tuned white-box LLM, AIO can still generally maintain the best performance compared with baseline methods. Similar to our zero-shot experiment settings, we average performance of AIO and its PEFT variant "AIO + LoRA" as a unity to obtain the ranking results.

| Tasks \ Methods | Black-box LLM | | White-box LLM | | White-box LLM w/ FT (Ours) | |
| --- | --- | --- | --- | --- | --- | --- |
| | APE | ProTeGi | InstructZero | INSTINCT | AIO | AIO + LoRA |
| antonyms | 0.901 | 0.889 | 0.894 | 0.905 | **0.912** | 0.895 |
| sentiment | 0.932 | 0.944 | 0.940 | 0.933 | **0.950** | 0.946 |
| larger_animal | 0.939 | 0.915 | 0.922 | 0.874 | **0.961** | 0.957 |
| taxonomy_animal | 0.708 | 0.972 | 0.835 | 0.869 | 0.967 | **0.976** |
| object_counting | 0.511 | **0.583** | 0.520 | 0.541 | 0.555 | 0.473 |
| navigate | 0.734 | 0.724 | 0.701 | 0.757 | **0.776** | 0.772 |
| winowhy | 0.563 | 0.674 | 0.673 | **0.682** | 0.628 | 0.621 |
| implicatures | 0.846 | 0.826 | 0.859 | 0.847 | 0.836 | **0.867** |
| logical_fallacy | 0.850 | **0.892** | 0.877 | 0.881 | 0.880 | 0.885 |
| hyperbaton | 0.556 | 0.595 | 0.580 | 0.641 | 0.634 | **0.660** |
| epistemic_reasoning | 0.712 | 0.802 | 0.622 | 0.765 | 0.774 | **0.884** |
| movie_recommendation | 0.930 | 0.948 | 0.955 | 0.960 | **0.979** | 0.963 |
| timedial | 0.760 | 0.820 | 0.748 | 0.784 | 0.779 | **0.832** |
| presuppositions_as_nli | 0.557 | 0.564 | 0.591 | 0.598 | **0.619** | 0.594 |
| question_selection | 0.879 | 0.882 | 0.781 | 0.822 | **0.916** | 0.887 |
| Average Rank | 4.20 | 2.73 | 3.67 | 2.67 | **1.73** | |

Table 3: Few-shot Instruction Induction Results. For each task (row), **bold** number refers to the best result, while underlined number refers to the second-best one. Similar to our zero-shot instruction induction experiments in Table 1, we average numerical results of AIO and its LoRA variant for each task, and treat these two methods as a unified baseline for ranking comparisons.

## B.4 CHAIN-OF-THOUGHT (COT) RESULTS

We also include additional Chain-of-Thought (CoT) experiments on three data sets including GSM8K (Cobbe et al., 2021), AQUA (Garcia et al., 2020), and SVAMP (Patel et al., 2021), where results are shown in Table 4. For comparison, we include a baseline instruction "Let's think carefully step by step", which is commonly applied for solving CoT tasks, following the settings from Chen et al. (2024); Lin et al. (2024).

| Data set | Method | Instruction | Accuracy Result |
| --- | --- | --- | --- |
| GSM8K | CoT | Let's think carefully step by step | 0.724 |
| | AIO | Use your math skills and logic to break down the problem into manageable parts. | 0.862 |
| AQUA | CoT | Let's think carefully step by step | 0.317 |
| | AIO | Think critically and break down the problem into smaller parts to solve it. | 0.410 |
| SVAMP | CoT | Let's think carefully step by step | 0.766 |
| | AIO | Let us think critically and break it down! | 0.898 |

Table 4: Chain-of-Thought (CoT) results.

With results in Table 4, we see that AIO can significantly improve black-box induction performance under CoT reasoning settings compared with the task-agnostic instruction "Let's think carefully step by step", where AIO's performance improvements can be credited to the utilization of task-aware instructions. For instance, since GSM8K is a math reasoning task, AIO choose to introduce additional background information by asking the task-solver black-box LLM to "use your math skills" and decompose the target math problem into "manageable parts". This can help the black-box LLM determine which part of or what kinds of learned knowledge should be applied for problem solving, with higher levels of clarity than task-agnostic instructions.

## B.5 Different combinations of white-box and black-box LLMs

Recall that for our previous experiments, we have applied `Llama-3-8B-Instruct` (Dubey et al., 2024) as our tunable white-box LLM $\mathcal{F}_W(\cdot; \Theta_W)$, and adopt `Claude-3-Sonnet` (Anthropic, 2024) as our black-box LLM $\mathcal{F}_B(\cdot)$. Here, for two tasks "navigate" and "larger animal", we include experiments with one more recent white-box LLM `Llama-3.1-8B-Instruct`, as well as a relatively light-weight black-box LLM `Claude-3-Haiku` for comparisons. Meanwhile, we also include experiments by substituting our black-box LLM with a powerful white-box LLM `Llama-3-70B-Instruct` for comparisons.

| Task | White-box LLM | Black-box LLM | Accuracy Result |
|------|---------------|---------------|-----------------|
| navigate | Llama-3-8B-Instruct | Claude-3-Sonnet | 0.644 |
| | Llama-3.1-8B-Instruct | Claude-3-Sonnet | 0.689 |
| | Llama-3-8B-Instruct | Claude-3-Haiku | 0.612 |
| | Llama-3.1-8B-Instruct | Claude-3-Haiku | 0.643 |

| Task | White-box LLM | Black-box LLM | Accuracy Result |
|------|---------------|---------------|-----------------|
| larger_animal | Llama-3-8B-Instruct | Claude-3-Sonnet | 0.912 |
| | Llama-3.1-8B-Instruct | Claude-3-Sonnet | 0.927 |
| | Llama-3-8B-Instruct | Claude-3-Haiku | 0.935 |
| | Llama-3.1-8B-Instruct | Claude-3-Haiku | 0.887 |

Table 5: Different combinations of white-box LLMs vs black-box LLMs.

| Task | White-box LLM | Task-solver LLM | Best Instruction | Accuracy Result |
|------|---------------|-----------------|------------------|-----------------|
| navigate | Llama-3-8B-Instruct | Llama-3-70B-Instruct | If the instructions are able to return to the starting position after following all the instructions, then the output is True. Otherwise, the output is False. | 0.567 |
| larger_animal | Llama-3-8B-Instruct | Llama-3-70B-Instruct | First identify the animals in the input. Then, sort the animals in descending order based on their average adult body mass. If there are multiple animals with the same average adult body mass, sort them in alphabetical order. Finally, return the first animal in the sorted list as the output. | 0.872 |

Table 6: Applying a white-box LLM (`Llama-3-70B-Instruct`) as problem-solving LLM.

Results are shown in Tables 5 and 6. Here, we see that using a more recent and capable white-box LLM can generally lead to slightly better performance. However, during our experiments, we also notice that fine-tuning `Llama-3.1-8B-Instruct` can be slightly more time consuming then tuning `Llama-3-8B-Instruct`. On the other hand, during our experiments, using a more light-weight black-box LLM can significantly accelerate the inference speed in terms of answer generation with relatively less API token costs. It can still achieve relatively good performance on "larger animal" task with a slightly inferior performance on "navigate" task. We also notice that the large white-box LLM `Llama-3-70B-Instruct` tends to perform slightly inferior compared with Claude family black-box LLMs, when using AIO as the instruction optimizer.

## B.6 Ablation Study on AIO Components

Recall that AIO has two main components: white-box back-propagation and TS-aided ZO gradient approximation, to derive the gradients for white-box LLM and black-box LLM respectively, based on the decomposed gradient flow in Eq. 4. Here, we include an ablation study for these two components for gradient derivation: (1) the first baseline is "AIO w/ MeZO" which directly use MeZO (Malladi et al., 2023) for approximating white-box LLM parameter gradients instead of our gradient decomposition formulation (Remark 1); (2) the second baseline is "AIO w/o TS Scheduling" where we do not apply Thompson Sampling for selecting perturbation directions and use completely random perturbation vectors $z \sim \mathcal{N}(0, I)$ for white-box gradient approximation in Eq. 5.

Experiment results are shown in Table 7. We can see that our proposed AIO with TS-aided ZO Gradient Approximation can still maintain superior performance compared with the other two baselines with substituted modules. This helps to reinforce our claim that our proposed gradient flow decomposition approach (Eq. 4) as well as the ZO black-box LLM gradient approximation method guided by Thompson Sampling are necessary for AIO to achieve optimal performance.

| Methods | antonyms | sentiment | l_animal | t_animal | navigate | implicatures | logical_fallacy | e_reasoning |
|---|---|---|---|---|---|---|---|---|
| AIO | **0.901** | **0.949** | **0.912** | **0.983** | 0.644 | **0.811** | **0.868** | **0.766** |
| AIO w/ MeZO | 0.852 | 0.930 | 0.847 | 0.279 | **0.654** | 0.675 | 0.812 | 0.748 |
| AIO w/o TS Scheduling | 0.870 | 0.929 | 0.760 | 0.957 | 0.604 | 0.787 | 0.832 | 0.742 |

Table 7: Ablation study on AIO with two variants: (1) "AIO w/ MeZO" directly applies MeZO for fine-tuning white-box LLM $\mathcal{F}_W$ instead of using our proposed gradient flow decomposition with TS-aided ZO gradient approximation; (2) "AIO w/o TS Scheduling" refers to a variant where perturbation directions $z$ are sampling randomly from $\mathcal{N}(0, I)$ instead of chosen by our TS model.

In particular, the supposed linear optimization landscape enables us to utilize a linear Thompson Sampling model for ZO perturbation direction selection, which is effective and computationally efficient for perturbation direction selection.

### B.7 PARAMETER STUDY FOR $\beta$ AND $B$

We include additional study for parameters $\beta$ and $B$ of AIO. Here, additional experiment results for zero-shot instruction induction, on the "Larger animal" and "Navigate" tasks with the LoRA module, are presented in the two tables below.

| Parameter $B$ | | | | |
|---|---|---|---|---|
| Task \ $B$ value | 1 | 2 | 3 | 4 |
| Larger Animal | 0.915 | 0.931 | 0.950 | 0.941 |
| Navigate | 0.610 | 0.632 | 0.627 | 0.664 |

Table 8: Experiment results with different $B$ values.

| Parameter $\beta$ | | | | |
|---|---|---|---|---|
| Task \ $\beta$ value | 1 | 10 | 100 | $\infty$ |
| Larger Animal | 0.922 | 0.915 | 0.932 | 0.950 |
| Navigate | 0.611 | 0.634 | 0.676 | 0.627 |

Table 9: Experiment results with different threshold $\beta$ values.

Results are shown in Tables 8 and 9. For parameter $B$, we observe that setting $B = 3$ can achieve promising performance, which can help balance computational (and token) costs with performance. For parameter $\beta$, it is recommended to start the tuning process with a large $\beta$, as suggested in our Appendix B.2, to effectively leverage past received records of the optimization landscape. On the other hand, a sufficiently small threshold $\beta$ will cause the method to degenerate into "AIO w/o TS Scheduling", as in our ablation study (Subsec. B.6). In this case, the TS model will be excluded from selecting ZO approximation directions, leading to relatively inferior performance.

### B.8 COMBINATIONS OF DIFFERENT WHITE-BOX LLMS

We include additional experiments with other types of white-box LLMs can benefit the audience. We conduct experiments with the LoRA module on two additional types of white-box LLMs: Mistral-7B-Instruct-v0.2 and Qwen2.5-7B-Instruct. With black-box LLM being Claude-3-Sonnet, we have zero-shot instruction induction results shown in Table 10. We see that our proposed AIO framework achieves promising performance with other types of white-box LLMs other than the Llama family. Meanwhile, we also would like to mention that Llama 3 (Llama-3-8B-Instruct) generally retains a slight advantage over the other two white-box LLMs. One possible reason is that Llama 3 consists of 8 billion parameters, slightly more than the other two 7-billion-parameter models. This can provide an advantage in terms of the representation power to some extent.

### B.9 DIFFERENT NUMBERS OF EXEMPLARS

As mentioned in our experimental settings (Appendix Subsec. B.2), we use 20 training exemplars, a reasonably small amount of training data, as the reference for the white-box LLM to generate

| Method \ Task | larger animal | navigate | sentiment | movie recommendation |
|---|---|---|---|---|
| Llama + Claude 3 | 0.950 | 0.627 | 0.947 | 0.883 |
| Mistral + Claude 3 | 0.890 | 0.634 | 0.907 | 0.874 |
| Qwen + Claude 3 | 0.919 | 0.682 | 0.922 | 0.835 |

Table 10: Experiment results with different kinds of white-box LLMs.

and optimize instructions. We also include additional zero-shot instruction induction experiments with varying exemplar quantities and the LoRA module. This is to investigate how the performance changes when altering the number of exemplars $|\mathcal{D}_{\text{Train}}|$ for AIO, in terms of instruction generation and optimization.

| Different numbers of exemplars $|\mathcal{D}_{\text{train}}|$ | | | | |
|---|---|---|---|---|
| Task \ $|\mathcal{D}_{\text{train}}|$ | 5 | 10 | 20 | 30 |
| Larger Animal | 0.868 | 0.921 | 0.950 | 0.947 |
| Navigate | 0.622 | 0.634 | 0.627 | 0.681 |

Table 11: Experiment results with different numbers of exemplars $|\mathcal{D}_{\text{train}}|$.

From the results in Table 11, we observe that the "larger animal" task can be more sensitive to the number of exemplars $|\mathcal{D}_{\text{Train}}|$. However, providing as few as $|\mathcal{D}_{\text{Train}}| = 10$ query-label pairs as exemplars, which is a modest quantity, will allow AIO to achieve relatively promising performance. Meanwhile, we see that for the "navigate" task, a small number of $|\mathcal{D}_{\text{Train}}| = 5$ exemplars can lead to promising results, which shows that it is more stable under the sparse data settings. Therefore, when fine-tuning AIO from scratch, the overall performance of AIO can possibly be influenced by data scarcity, with its impact varying based on the specific application scenarios of practitioners.

### B.10 TRANSFERABILITY OF FINE-TUNED INSTRUCTION OPTIMIZERS ACROSS DIFFERENT BLACK-BOX TASK-SOLVING LLMS

In this subsection, we investigate if the optimized instruction can generalize to different black-box LLMs. We transfer our optimized white-box LLM with the LoRA module, which is fine-tuned under the settings of Llama 3 + Claude 3 Sonnet, to other black-box task-solving LLMs. They include Claude 3.5 Sonnet and two OpenAI black-box LLMs (GPT-3.5-Turbo and GPT-4o).

| Method \ Task | epistemic | logical | hyperbaton | movie-recommendation |
|---|---|---|---|---|
| Claude 3 Sonnet | 0.719 | 0.836 | 0.527 | 0.883 |
| Claude 3.5 Sonnet | 0.844 | 0.886 | 0.562 | 0.924 |
| GPT-3.5-Turbo | 0.737 | 0.811 | 0.556 | 0.825 |
| GPT-4o | 0.768 | 0.854 | 0.540 | 0.910 |

Table 12: Experiment results with different kinds of black-box LLMs.

The accuracy results in terms of zero-shot instruction induction are shown in Table 12. Here, we can observe that the optimized instruction-generating white-box LLM can maintain strong performance when being applied to other black-box LLMs. Meanwhile, we also notice that the latest language models (Claude 3.5 and GPT-4o) can generally outperform the older ones (Claude 3 and GPT-3.5-Turbo), due to their stronger reasoning capabilities.

### B.11 ADDITIONAL BASELINE: EVOPROMPT

In this subsection, we included an additional baseline EvoPrompt (Guo et al., 2024), which alternatively utilizes evolutionary algorithms to refine LLM-generated instructions in an in-context learning manner. Different from our original problem settings, where instructions are generated from scratch, EvoPrompt requires initial instructions. In this case, we follow the settings in the original paper (Guo et al., 2024) and the official source code of EvoPrompt, by using instructions generated by APE (Zhou et al., 2022) as the initial instructions. We compare AIO with EvoPrompt, in terms of zero-shot instruction induction on four BigBench (bench authors, 2023) tasks, as BigBench is also utilized in (Guo et al., 2024).

| Method \ Task | hyperbaton | navigate | movie-recommendation | object-counting |
|---|---|---|---|---|
| AIO | 0.538 | 0.644 | 0.902 | 0.543 |
| EvoPrompt | 0.526 | 0.627 | 0.895 | 0.466 |

Table 13: Experiment results in comparison with EvoPrompt (Guo et al., 2024).

Based on the results in Table 13, we see that our proposed AIO can manage to achieve generally better performance compared with EvoPrompt. Meanwhile, comparing with the two black-box LLM baselines in our initial submission: APE (Zhou et al., 2022) and ProTeGi (Pryzant et al., 2023), EvoPrompt can achieve relatively better performance than them, by utilizing task prior knowledge provided by initial instructions as the starting point for optimization.

### B.12 ADDITIONAL INSTRUCTION OPTIMIZATION TRAJECTORY RESULTS

To provide additional insights for our instruction optimization process, apart from our instruction optimization trajectory results from Subsec. 5.3, we include complementary results on extra tasks, in terms of how generated instructions evolve over the fine-tuning process, as well as how their accuracy results change with the instruction optimization.

#### B.12.1 "SENTIMENT" TASK

- Accuracy: 46%

> IF the input contains any of the following words: "fun", "entertainingly", "splendid", "exquisite", "brilliantly", "impeccable", "likeable", "engrossing", "psychologically", "slyly" THEN output is positive.,
> ELSE IF the input contains any of the following words: "worst", "dislikable", "backlash", "mess", "pastiche", "warmed" THEN output is negative.

- Accuracy: 91%

> The output is positive if the input contains words with positive connotations, such as "fun", "entertainingly", "brilliantly", "exquisite", "likeable", "gentle", and "engrossing".
> The output is negative if the input contains words with negative connotations, such as "worst", "low", "pretentious", "dislikable", "mess", "warmed over", and "backlash".

#### B.12.2 "NAVIGATE" TASK

- Accuracy: 33%

> To determine if a robot will return to the starting point after a series of movements, follow these steps:
> 1. Count the total number of steps taken by the robot.
> 2. If the robot turns around at any point, it will change direction. So, count the number of turns and subtract 2 from the total number of steps for each turn.
> 3. If the robot always faces forward, it will not change direction. So, do not count the turns.
> 4. If the total number of steps minus the adjustments for turns is a multiple of the initial number of steps, the robot will return to the starting point. Example: If the robot takes 10 steps forward and then 5 steps backward, the total number of steps is 15. If the robot turns around at some point, adjust the total number of steps by subtracting 2 for each turn. If the robot always faces forward, do not adjust the total number of steps. If the total number of steps minus the adjustments is a multiple of the initial number of steps, the robot will return to the starting point.

- Accuracy: 71%

To determine if a robot will return to its initial position after a series of instructions, follow these steps:
1. Initialize the robot's position and direction to the starting point.
2. Iterate through the instructions: - If the instruction is "Turn around", update the robot's direction to the opposite direction. - If the instruction is "Turn right" or "Turn left", update the robot's direction accordingly. - If the instruction is "Take x steps", update the robot's position by moving x steps in the current direction.
3. After iterating through all instructions, check if the robot's final position is the same as the initial position.
4. If the final position is the same as the initial position, the output is True; otherwise, the output is False.

### B.12.3   "OBJECT COUNTING" TASK

- Accuracy: 27%

To infer the output for a given input, count the number of items in the input that are not fruits or vegetables. Then, return the count. For example, if the input is "I have a violin, a piano, an accordion, a clarinet, an apple, a trumpet, and a banana.", you would count the number of non-food items, which are 7 (violin, piano, accordion, clarinet, trumpet), and return 7. If the input is "I have a yam, a carrot, a cabbage, two garlics, a potato, a stalk of celery, a head of broccoli, and a cauliflower.", you would count the number of non-food items, which are 0, and return 0.

- Accuracy: 52%

To infer the output for a given input, follow these steps:
1. Identify the types of items in the input. For example, if the input contains "violin", "piano", "accordion", "clarinet", "apple", "trumpet", and "banana", the types of items are musical instruments and fruits.
2. Count the number of items of each type. For example, there is 1 violin, 1 piano, 1 accordion, 1 clarinet, 1 apple, 1 trumpet, and 1 banana.
3. Add the counts of all types of items together to get the total number of items.
4. The output is the total number of items.
For example, if the input is "I have a violin, a piano, an accordion, a clarinet, an apple, a trumpet, and a banana.", the output would be 7.

### B.12.4   "TAXONOMY ANIMAL" TASK

- Accuracy: 0%

Sort the input list in descending order based on the length of the words. Then, select the words that are animals. Finally, sort the remaining words in descending order based on the length of the words.

- Accuracy: 95%

1. First, identify all the words that are not animals. These words are usually nouns that are not animals, such as food items, colors, objects, etc.
2. Sort the remaining animal words in alphabetical order.
3. The output is the list of animal words in alphabetical order, excluding the non-animal words.

### B.12.5   "LARGER ANIMAL" TASK

- Accuracy: 50%

> 1. Take the first animal from the input.
> 2. If the first animal is a predator, compare it with the second animal.
> 3. If the first animal is a predator and the second animal is not a predator, then the output is the second animal.
> 4. If the first animal is not a predator, compare it with the second animal.
> 5. If the first animal is not a predator and the second animal is a predator, then the output is the first animal.
> 6. If neither of the animals is a predator, compare their sizes.
> 7. If the first animal is larger than the second animal, then the output is the first animal.
> 8. If the first animal is smaller than the second animal, then the output is the second animal.
> 9. If the first animal and the second animal are of the same size, then the output is the first animal.

- Accuracy: 92%

> If the input consists of two animals, compare their sizes. If the first animal is larger, output the first animal. If the first animal is smaller, output the second animal. If both animals have the same size, output the first animal.

### B.12.6 "IMPLICATURES" TASK

- Accuracy: 62%

> Given two speakers' dialogue, analyze the context and sentiment of the speakers' statements to determine if the second speaker is thanking the first speaker (output: 'yes') or not (output: 'no'). To do this, consider the following factors:
> 1. **Gratitude tone**: If the second speaker's statement conveys gratitude, appreciation, or a positive sentiment towards the first speaker, infer 'yes'.
> 2. **Question-answer format**: If the first speaker's statement is a question and the second speaker's response is an affirmative answer or a statement that implies agreement, infer 'yes'.
> 3. **Contrast**: If the first speaker's statement is a negative or neutral statement, and the second speaker's response is a contrasting positive statement, infer 'yes'.
> 4. **Contextual understanding**: Consider the context of the conversation and the speakers' intentions. If the second speaker's statement seems to be responding to the first speaker's action or suggestion in a positive manner, infer 'yes'.
> When applying these factors, consider the nuances of language, idioms, and figurative expressions. If the analysis is ambiguous or unclear, default to 'no'.

- Accuracy: 84%

> To infer the output for a given input, follow these steps:
> 1. Identify the speaker's tone and intention in the input. Are they expressing gratitude, surprise, or skepticism?
> 2. Look for words or phrases that convey a positive or negative sentiment. If the sentiment is positive, the output is likely to be 'yes'. If the sentiment is negative, the output is likely to be 'no'.
> 3. Check for words or phrases that indicate a question or a request. If the input contains a question, the output is likely to be 'yes' if the speaker is seeking confirmation or agreement. If the input contains a request, the output is likely to be 'yes' if the speaker is seeking permission or approval.
> 4. Consider the context of the conversation. Is the speaker discussing a specific event, situation, or decision? If so, the output may be influenced by the speaker's perspective or attitude towards the topic.
> 5. Use your judgment and common sense to make an educated guess about the output. If you're still unsure, try re-reading the input and re-evaluating the tone, sentiment, and context.
> By following these steps, you can make an informed decision about the output for a given input.

### B.12.7 "EPISTEMIC REASONING" TASK

- Accuracy: 41%

> For each input, check if the premise and hypothesis are about the same individual or group of individuals. If they are, the output is entailment. If they are not, the output is non-entailment.

- Accuracy: 76%

> Given a premise and a hypothesis, determine the entailment relationship between them. For a premise, identify the main clause and the subordinate clause (if present). For the hypothesis, identify the main clause and the subordinate clause (if present). If the main clause in the premise is identical to the main clause in the hypothesis, and the subordinate clause (if present) is also identical, then the output is entailment. If the main clause in the premise is identical to the main clause in the hypothesis, but the subordinate clause (if present) is different, then the output is non-entailment. If the main clause in the premise is not identical to the main clause in the hypothesis, then the output is non-entailment.

### B.12.8 "TIMEDIAL" TASK

- Accuracy: 41%

> 1. Read the context carefully and identify the main idea or topic of the conversation.
> 2. Identify the speaker and the listener.
> 3. Determine the purpose of the conversation.
> 4. Identify the key information or events mentioned in the conversation.
> 5. Analyze the conversation to determine the relationship between the speakers and the context.
> 6. Based on the analysis, infer the output for the given input.

- Accuracy: 79%

> To infer the output for a given input, follow these steps:
> 1. Identify the context and the entities mentioned in the input, such as people, places, and times.
> 2. Look for keywords and phrases that indicate the type of information being requested or discussed, such as dates, times, quantities, or specific events.
> 3. Analyze the relationships between the entities and the context, and try to identify the most relevant and important information.
> 4. If the output is a date or time, check if it is a specific date or time, or if it is a range or duration.
> 5. If the output is a quantity, check if it is a specific number, or if it is a range or interval.
> 6. Finally, use the inferred output to generate a response that is relevant and accurate to the input and the context.

### B.12.9 "QUESTION SELECTION" TASK

- Accuracy: 39%

> To infer the output for a given input, follow these steps:
> 1. Identify the context of the input, which typically consists of a passage of text.
> 2. Determine the type of question being asked, which is usually indicated by the choice options provided.
> 3. Look for specific keywords or phrases in the input that are related to the question being asked.
> 4. Identify the relevant information in the input that answers the question, which may be a specific fact, statistic, or quote.
> 5. Match the relevant information to the corresponding choice option in the question.
> 6. Select the choice option that best answers the question based on the information provided in the input.

- Accuracy: 64%

> To infer the output for a given input, follow these steps:
> 1. Identify the context of the input, which is typically a passage of text.
> 2. Determine the type of question being asked, which is usually indicated by the format of the choices provided.
> 3. Scan the context to identify relevant information, such as key phrases, names, and dates.
> 4. Match the context to the corresponding choices, considering the question type and the information gathered.
> 5. Select the most likely output based on the context and the choices.
> For example, if the input is a passage about a football game, the output might be a question about the teams involved, the score, or the MVP. If the input is a passage about a historical event, the output might be a question about the date, location, or significance of the event.

### B.12.10  "MOVIE RECOMMENDATION" TASK

- Accuracy: 48%

> 1. Identify the most frequent genre or theme in the given context.
> 2. Compare the given context with the choices and find the one that best matches the identified genre or theme.
> 3. Output the matching choice.

- Accuracy: 88%

> The output is the choice that is most commonly associated with the given context, based on the frequency of co-occurrences of movies in the input list with each choice.

## C  SUPPLEMENTARY TECHNICAL DETAILS

### C.1  DETAILS FOR AUTO-REGRESSIVE GENERATION AND PERTURBATION

Analogous to existing works (e.g., Li & Liang (2021)), we formulate the LLM generation process of a token sequence $x$. Beginning with an initial input context $x_{<1}$ that can be empty or contain special tokens like start-of-sequence token. Then, the language model will sequentially generate each token in the output, by sampling each generated token $x_i$ from the conditional distribution $p_{\Theta}(x_i \mid x_{<i})$. In particular, the probability distribution for the $i$-th token will go through a softmax function, after applying a language model header (with weight $\Theta_{\text{header}}$ from language model parameters $\Theta$) to map $i$-th token hidden representation to the vocabulary distribution, as

$$p_{\Theta}(x_i \mid x_{<i}) = \text{softmax}(\Theta_{\text{header}} \cdot h_i),$$

where $h_i$ represents the transformer-embedded hidden representation of $i$-th generated token. Each token $x_i$ will be sampled from the vocabulary, based on $p_{\Theta}(x_i \mid x_{<i})$. The generation process will complete if special tokens (e.g., an EOS token) is encountered, or the maximum length is met.

Following our gradient approximation formulation in Eq. 5, with random perturbation vector (gradient approximation vector) $\boldsymbol{z} \sim \mathcal{N}(0, \boldsymbol{I})$ and perturbation magnitude $\epsilon > 0$, we recall that the perturbation is imposed on *LLM-header output probabilities* (i.e., distribution over the vocabulary) of *each $i$-*th generated token. This leads to the positively perturbed generation process $p_{\boldsymbol{\Theta}}(x_i \mid \boldsymbol{x}_{<i}) \leftarrow$ softmax($\boldsymbol{\Theta}_{\text{header}} \cdot \boldsymbol{h}_i + \epsilon \boldsymbol{z}$), as well as the negatively perturbed generation process $p_{\boldsymbol{\Theta}}(x_i \mid \boldsymbol{x}_{<i}) \leftarrow$ softmax($\boldsymbol{\Theta}_{\text{header}} \cdot \boldsymbol{h}_i - \epsilon \boldsymbol{z}$). Consequently, the perturbation $\boldsymbol{z} \in \mathbb{R}^d$ will be of the same as the vocabulary dimension $d$.

## C.2 DETAILS FOR EFFICIENT COVARIANCE MATRIX INVERSION UPDATE WITH SHERMAN-MORRISON FORMULA

Recall that in Remark 2, for updating covariance matrix inversion $\boldsymbol{\Sigma}_t^{-1}$ efficiently in each round $t$, we apply Sherman-Morrison Formula (Bartlett, 1951; Maponi, 2007) by updating matrix inversion incrementally. Here, suppose that current white-box LLM output $\phi$ is close enough to the checkpoint $\phi_{\text{Check}}$, which means that we can update currently possessed covariance matrix inversion $\boldsymbol{\Sigma}_{t-1}^{-1}$ to obtain $\boldsymbol{\Sigma}_t^{-1}$. Then, we recall that the arm context covariance matrix in each round $t$ is constructed by $\boldsymbol{\Sigma}_t = \boldsymbol{I} + \sum_{(\boldsymbol{z},r) \in \Omega_t} \boldsymbol{z} \cdot \boldsymbol{z}^{\intercal} = \boldsymbol{\Sigma}_{t-1} + \sum_{\boldsymbol{z} \in \widetilde{\mathcal{Z}}_t} \boldsymbol{z} \cdot \boldsymbol{z}^{\intercal}$, where $\widetilde{\mathcal{Z}}_t$ refers to the collection of chosen arms in round $t$. Since we have each $\boldsymbol{z}\boldsymbol{z}^{\intercal}$ being a rank-one matrix for every $\boldsymbol{z} \in \widetilde{\mathcal{Z}}_t$, we can follow Sherman-Morrison Formula to perform one-step update

$$\left(\boldsymbol{\Sigma}_{t-1} + \boldsymbol{z}\boldsymbol{z}^{\intercal}\right)^{-1} = \boldsymbol{\Sigma}_{t-1}^{-1} - \left[\frac{\boldsymbol{\Sigma}_{t-1}^{-1}\boldsymbol{z}\boldsymbol{z}^{\intercal}\boldsymbol{\Sigma}_{t-1}^{-1}}{1 + \boldsymbol{z}^{\intercal}\boldsymbol{\Sigma}_{t-1}^{-1}\boldsymbol{z}}\right].$$

In this case, by iteratively repeating this process for $|\widetilde{\mathcal{Z}}_t| = B$ times (since we have $B \ll K$ chosen arms in each round $t$, and $B$ is a considerably small integer), we will have the updated covariance matrix inverse $\boldsymbol{\Sigma}_t^{-1}$. Recall that each covariance matrix will have a shape of $d \times d$, where $d$ is the dimensionality of perturbation vector $\boldsymbol{z}$. We then have the overall computational costs as approximately $\mathcal{O}(Bd^2)$ instead of the naive $\mathcal{O}(d^3)$, where we also intuitively have $B \ll d$. Moreover, with dimension reduction approach motivated by JL-Lemma (Remark 2), we can have the projected context dimension $d' \ll d$, which leads to computational complexity of $\mathcal{O}(B \cdot (d')^2)$, instead of the naive $\mathcal{O}((d')^3)$ with the direct matrix inversion.

# D COMPLEMENTARY DISCUSSIONS

## D.1 MOTIVATION OF THOMPSON SAMPLING AND BIAS ASSOCIATED WITH THIS DESIGN

### D.1.1 ADDITIONAL DISCUSSION ON MOTIVATIONS

As mentioned in our manuscript (paragraph below Remark 1), conventional ZO gradient approximation methods have a potential drawback: the perturbation vectors $\boldsymbol{z}$ are randomly sampled. Consequently, their gradient approximation directions in the optimization landscape are random, which can result in inefficient gradient estimation process. To overcome this challenge, numerous ZO methods have been proposed to incorporate guided or chosen directions for gradient optimization, enabling more efficient estimation of the target gradient (e.g., Cai et al. (2022); Qiu & Tong (2024)).

In this work, we propose reusing collected feedback by framing the ZO-based fine-tuning process as an online sequential decision-making problem, and applying Contextual Bandit techniques to effectively identify beneficial perturbation directions worth exploring. With the assumed linear optimization landscape as in existing works (e.g., Spall (1992); Malladi et al. (2023)), we apply linear Thompson Sampling to leverage previously collected information, which includes arms (previous gradient approximation directions) and corresponding rewards (benefits of going along these directions), to achieve a more efficient gradient estimation (Subsec. 4.1.2). Within the linear optimization landscape, our TS model aids in selecting gradient directions for white-box LLM fine-tuning, by properly reducing the possibility of choosing low-value directions (e.g., directions that are orthogonal to the true gradient, which can provide limited information for gradient approximation). This approach helps reduce the validation loss through instruction generation and ZO approximation direction selection, aligned with our formulation of the arm reward (Eq. 8).

**Ablation study on AIO components.** We also would like to mention that the effectiveness of our TS modeling is supported by our ablation study (Appendix Subsec. B.6), where we compare our AIO framework with two alternatives: (1) "AIO w/ MeZO", where MeZO (Malladi et al., 2023) is used directly for approximating white-box LLM parameter gradients instead of our gradient decomposition formulation (Remark 1). (2) "AIO w/o TS Scheduling", where Thompson Sampling is not applied for selecting perturbation directions, and completely random perturbation vectors $z \sim \mathcal{N}(0, \boldsymbol{I})$ are used for zeroth-order gradient approximation as in Eq. 5. We observe that our proposed AIO with TS-aided ZO gradient approximation generally achieves better performance, compared with the two baselines with substituted modules.

### D.1.2 BIAS ASSOCIATED WITH TS

*By balancing exploitation and exploration, TS will not introduce considerable bias.* Under the settings of gradient approximation, our proposed TS-based zeroth-order gradient approximation method itself does not inherently introduce considerable bias in terms of gradient estimation, as Thompson Sampling techniques can naturally tackle the exploitation-exploration dilemma (Agrawal & Goyal, 2013; Zhang et al., 2021) by exploring various gradient directions instead of greedily exploiting only certain ones. Here, TS model will choose from sampled candidate approximation directions, instead of actually altering the direction vector value. Meanwhile, in the short term, it is possible that TS may introduce temporary bias if it focuses on some perturbation directions that can lead to high rewards (i.e., directions that can help reduce validation loss, Eq. 8). Due to the *bias-variance trade-off* (Hu et al., 2021b), the temporary bias can also be beneficial under our gradient approximation settings, and we will elaborate on this point in the next paragraph. On the other hand, with only a few gradient directions applied for LLM fine-tuning (Malladi et al., 2023) in each optimization round, directly applying conventional unbiased estimators with randomly sampled directions can possibly result in high variance in terms of gradient estimation (Cai et al., 2022), which also reflects the *bias-variance trade-off* in terms of zeroth-order gradient approximation.

*We introduce the TS to balance the "bias" and "variance" trade-off for ZO gradient approximation, under query-limited scenarios.* Here, we would like to mention that state-of-the-art ZO gradient approximation methods with a guided approximation process (e.g., Qiu & Tong (2024)) can also introduce temporary bias in terms of gradient direction selection. However, they have been shown highly effective and efficient, particularly under sample-efficient settings, due to their ability of involving informative gradient approximation directions to balance "bias" and "variance" for gradient approximation. As a result, due to the bias-variance trade-off, it can lead to the case that *no algorithm can serve as a universal solution to various application scenarios of zeroth-order approximation*. Therefore, practitioners need to tailor solutions to their specific application scenarios. In our case, by balancing high-reward directions (exploitation) and sampling TS parameters from the posterior (exploration), our TS-aided approximation can help stabilize the gradient estimates, making it advantageous in query-limited scenarios like ours. In particular, by leveraging a highly efficient linear TS model, the arm selection process ($\sim 0.4$ seconds per round) will be fast, ensuring the efficiency in terms of gradient direction selection.

**Ablation study on TS-aided zeroth-order gradient approximation.** The effectiveness of our modeling is also validated by our ablation study (Appendix Subsec. B.6), where we include a variant, "AIO w/o TS Scheduling," for comparison. Instead of using TS, "AIO w/o TS Scheduling" applies randomly chosen perturbation vectors $z \sim \mathcal{N}(0, \boldsymbol{I})$ for zeroth-order gradient approximation, as described in Eq. 5. In contrast, our AIO with TS-aided ZO gradient approximation achieves better performance by strategically selecting informative gradient approximation directions, based on collected information and knowledge from the optimization landscape.

### D.2 REASONING OF OUR CHOICE OF DATA SETS AND BASELINES

Under our problem settings (Section 2), the black-box LLM is considered as part of the learning objective rather than the learning model, and we have no control over its parameters while can only interact with it through the API. Therefore, based on our problem settings, which align with those of closely related works (Chen et al., 2024; Lin et al., 2024), the learning model will solely be the white-box LLM, while the black-box LLM will serve as part of the learning objective.

### D.2.1 CHOICE OF DATA SETS.

Different from conventional instruction optimization settings, in this work, we address the challenge of automatically optimizing instructions for a task-solving black-box LLM in terms of the given target task. The process requires only a few exemplars as training data and eliminates the need for human expert intervention. This is an emerging topic of automatic instruction generation that has been explored in several related works (e.g., Zhou et al. (2022); Chen et al. (2024); Lin et al. (2024)). Unlike existing approaches, we propose utilizing a white-box LLM for instruction generation and optimization, coupled with LLM fine-tuning to effectively learn optimized instructions for highly complex modern black-box LLMs.

As discussed in the Introduction and Related Works sections, the most relevant state-of-the-art works to this paper are Chen et al. (2024); Lin et al. (2024), where a white-box LLM is also used as an instruction optimizer to tailor instructions specifically for the downstream task-solving black-box LLM. Therefore, for the datasets, we follow the most closely related works (e.g., Zhou et al. (2022); Chen et al. (2024); Lin et al. (2024)) by utilizing instruction induction tasks from Honovich et al. (2022), reasoning tasks from BigBench (bench authors, 2023), and Chain-of-Thought (CoT) datasets (Cobbe et al., 2021; Garcia et al., 2020; Patel et al., 2021) to benchmark our proposed AIO against baselines. These datasets are both common and widely adopted in this line of research, particularly in works closely aligned with ours in terms of problem settings (e.g., Zhou et al. (2022); Chen et al. (2024); Lin et al. (2024)).

The instruction induction tasks from Honovich et al. (2022) and reasoning tasks from BigBench (bench authors, 2023) are used to evaluate the zero-shot instruction induction (Subsec. 5.1) and few-shot instruction induction (Subsec. B.3) quality of the optimized instructions. In particular, since no auxiliary task-relevant information is provided to the task-solving black-box LLM, the zero-shot instruction induction results on these datasets (Honovich et al., 2022; bench authors, 2023) are widely adopted to assess instruction quality by closely related works (e.g., Zhou et al. (2022); Chen et al. (2024); Lin et al. (2024); Fernando et al. (2023); Hu et al. (2024)). Additionally, the applied CoT datasets are also commonly used in the aforementioned related works (e.g., Zhou et al. (2022); Chen et al. (2024); Lin et al. (2024)), as they effectively test the instruction optimizers' ability to solve complex reasoning tasks, such as math problems. Therefore, our dataset selections are standard and widely adopted in closely related works, in order to demonstrate the effectiveness of AIO in terms of instructing the task-solving black-box LLM.

### D.2.2 CHOICE OF BASELINES.

In this paper, we primarily focus on the challenge of automatically optimizing instructions for a task-solving black-box LLM given the target task. This distinguishes our problem settings from those of conventional instruction optimization works. In this case, the most relevant works to ours are Chen et al. (2024); Lin et al. (2024), which propose to deal with an analogous problem. From an in-context learning perspective instead, they apply a white-box LLM as an instruction generator to tailor instructions specifically for the downstream task-solving black-box LLM. Therefore, in our initial submission, we have included these two works (InstructZero (Chen et al., 2024), INSTINCT (Lin et al., 2024)) as our baselines to emphasize the advantages of LLM fine-tuning over their in-context learning approaches under our instruction optimization settings, given the superior representation and learning capabilities provided by LLM fine-tuning.

Meanwhile, there is also a line of research (e.g., Zhou et al. (2022); Pryzant et al. (2023)) that employs a black-box LLM instead for instruction generation, resulting in a pipeline involving two black-box LLMs. In this setup, the instruction-generating black-box LLM perceives the feedback of the task-solver black-box LLM, and optimizes its instructing strategy from an in-context learning perspective. Since these methods are also commonly adopted by our closely related works (Chen et al., 2024; Lin et al., 2024), we have also included APE (Zhou et al., 2022) and ProTeGi (Pryzant et al., 2023) as our baselines in our initial submission. Therefore, our choice of baselines is standard in this line of research, including state-of-the-art baselines closely related to this paper, as well as common baselines adopted by other existing works in this research direction.

# E  GENERALIZING AIO TO OTHER APPLICATION SCENARIOS

## E.1  INVOLVING DOMAIN EXPERTS IN THE OPTIMIZATION OF INSTRUCTIONS

As shown in our pipeline illustration (the Generation Template), we only need a few exemplars from the target task as the reference to the white-box LLM for instruction generation and optimization. In this case, we do not need specific human-crafted task context for our AIO, which avoids the need for human expert intervention. This emerging topic in automatic instruction generation has been explored in several related works (e.g., Zhou et al. (2022); Chen et al. (2024); Lin et al. (2024)). Unlike prior approaches, we propose utilizing a white-box LLM for instruction generation and optimization, paired with LLM fine-tuning to efficiently learn optimized instructions for modern, highly complex black-box LLMs. On the other hand, if domain experts (e.g., human engineers) or domain knowledge (e.g., textual task descriptions or narratives) are available, there are several ways of integrating them to our instruction optimization process.

### E.1.1  TASK NARRATIVE OR DESCRIPTION

First, if task descriptions are provided to AIO as prior knowledge, we can incorporate this information into the instruction generation template (left-hand side of Figure 1). In this case, the input to the white-box LLM will include both the task exemplars and the textual task description. We conduct additional experiments in terms of zero-shot instruction induction on four BigBench tasks with the LoRA module, incorporating the one-sentence task descriptions provided by the BigBench data set into our generation template.

| Method \ Task | epistemic reasoning | logical fallacy | hyperbaton | movie recommendation |
|---|---|---|---|---|
| AIO | 0.719 | 0.836 | 0.527 | 0.883 |
| AIO w/ Task Info | 0.811 | 0.872 | 0.639 | 0.895 |

Table 14: Experiment results with additional textual task description information.

Results are shown in Table 14. We see that involving additional textual task information into the instruction generation and optimization process can generally improve performance, particularly for logical reasoning tasks (e.g., epistemic reasoning, logical fallacy, hyperbaton), as task descriptions can help provide extra background information to prevent potential misinterpretations of exemplars.

### E.1.2  INVOLVING HUMAN EXPERTS

When human experts are available, following the idea of "Human-in-the-loop" (Wu et al., 2022), we can involve these experts to help AIO in terms of instruction optimization and evaluation. For instance, in terms of instruction improvement for medical diagnostics, AIO can generate instructions such as "Please analyze patient symptoms to identify possible illnesses." Human evaluators could define metrics emphasizing clarity (e.g., suggesting tools like "blood test" or "MRI scan") or appropriateness (e.g., ensuring instructions do not encourage unsafe practices). This can enhance AIO's applicability in sensitive fields where clarity and accuracy are crucial. Compared to composing instructions from scratch that requires more efforts, this formulation can help reduce the workload of human experts.

Meanwhile, human experts can be involved into the instruction evaluation process. In this case, AIO can be requested to generate multiple candidate instructions after fine-tuning, and then human experts can proceed to examine which of the candidates is most appropriate from multiple dimensions, such as interpretability. This will also help to enhance AIO's applicability for real-world scenarios.

## E.2  GENERALIZING OPTIMIZED INSTRUCTION TO RELATED TASKS OR DOMAINS

Recall that this work focuses on the problem of automatically optimizing instructions for a task-solving black-box LLM given the target task. The instruction generation and optimization require only a few exemplars as training data and do not involve human expert intervention. This problem is an emerging topic in the field of automatic instruction generation, with several related works (e.g., Zhou et al. (2022); Chen et al. (2024); Lin et al. (2024)). Different from existing works, in this paper, we propose leveraging a white-box LLM for instruction generation and optimization, combined with LLM fine-tuning to effectively learn optimized instructions for modern black-box LLMs of extreme

complexity. However, while this paper focuses on optimizing instructions for a given target task, our AIO framework can be potentially extended to generalize across multiple related tasks or domains.

**Transfer learning and domain adaptation.** First, we can leverage ideas from transfer learning and domain adaptation to address this issue. Here, we can consider two related domains (tasks): the source domain and the target domain. The source domain will refer to the domain that we train the original instruction optimizer (white-box LLM), typically having larger amount of training data (exemplars) than the target domain. Afterwards, to adapt the trained white-box LLM to the target domain, we can apply few-shot domain adaptation techniques (e.g., Chronopoulou et al. (2022)) to efficiently fine-tune the white-box LLM parameters for the target domain, even with a relatively smaller number of samples. In the meantime, in the context of transfer learning, we can possibly have additional theoretical insights. This is because the generalization loss on the target domain can be theoretically upper bounded by the empirical loss on the source domain, the learning power of the neural model, and the discrepancy between the source and target domains (Ben-David et al., 2010).

**Meta-learning.** On the other hand, we can leverage ideas from meta-learning (Finn et al., 2017) to develop an instruction "meta-optimizer", which is capable of abstracting high-level information across multiple domains and generating high-quality meta-instructions (analogous to meta-prompts (Hou et al., 2022)). Using this approach, the "meta-optimizer" can be adapted to downstream tasks with only a few exemplars from the target task, without significant modifications of the AIO parameters. The performance of this approach is expected to depend on the neural model's learning capacity and the discrepancy among different tasks (Chen et al., 2021). With a white-box LLM as the learning model with fine-tuning, providing sufficient representation power, AIO can help acquire high-level instruction knowledge using meta-learning techniques.

