# OpenReview forum: "Automatic Task-aware Instruction Optimizer for Black-box LLMs"
_ICLR.cc/2025/Conference — Submitted to ICLR 2025_

### Official Review · Reviewer_f3S6 · 2024-10-25

**Soundness:** 3
**Presentation:** 3
**Contribution:** 3
**Rating:** 6
**Confidence:** 3

**Summary:**

This paper mainly focuses on optimizing task-aware instructions in the absence of task-solver LLMs’ internal parameters and gradient information. Finding that the current instruction optimization methods based on in-context learning are insufficient to match the extreme complexity of modern black-box LLMs, this paper proposes a novel framework, called Automatic Instruction Optimizer (AIO), which fine-tunes a white-box LLM and enables it to adaptively adjust its instructing strategy for a particular task-solver black-box LLM. To address the optimization challenges, AIO adopts zeroth-order gradient approximation and Contextual Bandit techniques, eliminating the need for internal parameters and gradient information.

**Strengths:**

- This paper demonstrates that fine-tuning a white-box LLM into the instruction optimizer is able to have better instruction generation performance due to a stronger representation power.
- This paper addresses the optimization challenges using zeroth-order gradient approximation and manages to reduce the API cost by reusing historical records.
- It conducts comprehensive experiments to verify the effectiveness of the proposed method, including different combinations of white-box and black-box LLMs, instruction optimization trajectory results, comparison to CoT.

**Weaknesses:**

- A minor point: study the transferability of fine-tuned instruction optimizers across different black-box task-solver LLMs.
- The right-hand side of equation 1 misses a right parenthesis.
- Figure 3 lacks proper legends for different instruction optimization methods.

**Questions:**

Kindly refer to the weaknesses.

---

> ### Author Response · Authors · 2024-11-20
> **Thank you for the insightful review**
>
> We sincerely thank the reviewer for your valuable questions and comments, which will certainly help improve the paper's presentation and make it more solid.
> We will provide a detailed response in the form of Q\&A. If you have any further questions or comments, please feel free to let us know. We would be happy to provide additional responses promptly. **Thank you!**
>
>
>
>
> ### Q1: Transferability of fine-tuned instruction optimizers across different black-box task-solving LLMs?
>
>
> We agree with the reviewer that it will be beneficial to investigate whether the optimized instruction can generalize to different black-box LLMs.
> We transfer our optimized white-box LLM with the LoRA module, which is fine-tuned under the settings of Llama 3 + Claude 3 Sonnet, to other black-box task-solving LLMs. They include Claude 3.5 Sonnet and two OpenAI black-box LLMs (GPT-3.5-Turbo and GPT-4o).
>
>
>
>
> | Black-box LLM $\backslash$ Task |  epistemic reasoning |  logical fallacy | hyperbaton | movie recommendation
> |---  |--- |--- |--- |--- |
> |  Claude 3 Sonnet |   0.719    |   0.836  |   0.527  |   0.883
> |  Claude 3.5 Sonnet   |   0.844  |   0.886   | 0.562 |   0.924
> | GPT-3.5-Turbo   |   0.737  |   0.811   | 0.556 |   0.825
> | GPT-4o   |   0.768  |   0.854   | 0.540 |   0.910
>
>
> **Experiment results with different kinds of black-box LLMs.**
>
>
>
>
> The accuracy results in terms of zero-shot instruction induction are shown in the table above.
> Here, we can observe that the optimized instruction-generating white-box LLM can maintain strong performance when being applied to other black-box LLMs. Meanwhile, we also notice that the latest language models (Claude 3.5 and GPT-4o) can generally outperform the older ones (Claude 3 and GPT-3.5-Turbo), due to their stronger reasoning capabilities.
>
>
>
> ### Q2: The right-hand side of equation 1 misses a right parenthesis? Figure 3 lacks legend?
>
>
> We sincerely thank the reviewer for spotting these typos. We have properly updated the manuscript to fix these issues.
> *For your reference, please kindly refer to the updated manuscript that has been uploaded to the OpenReview platform.*

---

> ### Author Response · Authors · 2024-11-27
> **Looking forward to your feedback**
>
> Dear Reviewer f3S6,
>
> Thank you once again for your insightful review, comments, and questions, which have been crucial in improving the quality of our work.
> **As the PDF update will be disabled tomorrow, we kindly ask if our responses have adequately addressed your concerns regarding our work.**
> We would be sincerely grateful if you could review our responses and let us know if they sufficiently address your concerns. Any further comments or suggestions you may have would be greatly valued.
> **The current rebuttal discussions and additional empirical results have been incorporated into the manuscript**. We will continue to update the manuscript if you have any further comments or suggestions.
>
> For the response to your questions and concerns:
>
> 1. In Q1, as suggested by the reviewer, we included additional experimental results on **transferring fine-tuned instruction optimizers across different black-box LLMs** (Claude 3.5 Sonnet, GPT-3.5 Turbo, GPT-4o).
>
> 2. In Q2, we addressed the issue of the missing legend and typo in our initial manuscript.
>
>
>
>
>
> **We kindly invite you to refer to our updated manuscript for your reference.** Thank you once again for your time and dedication to the review process.
>
>
>
> Sincerely,
>
> Authors

---

> ### Author Response · Authors · 2024-12-02
> **Final day of the discussion phase: Please kindly check our response.**
>
> Dear Reviewer f3S6,
>
> We would like to sincerely thank you again for your time and effort in reviewing our submission. Your feedback has been invaluable in improving the quality of our work, and we have carefully revised our manuscript as well as uploaded the latest version based on your comments.
>
> As tomorrow is the final day of the author-reviewer discussion phase, we kindly ask if you could review our responses to your comments and let us know if they address your concerns. We would greatly appreciate any additional feedback or suggestions you may have.
>
> Thank you again for your dedication to the review process, and we look forward to hearing from you.
>
> Best regards,
>
> Authors

---

### Official Review · Reviewer_r8CJ · 2024-10-26

**Soundness:** 2
**Presentation:** 2
**Contribution:** 2
**Rating:** 5
**Confidence:** 4

**Summary:**

This paper proposes a task-aware instruction optimization for black-box LLMs. This method leverages both white LLM and black LLM. AIO adopts the parameter fine-tuning process with gradient approximation and contextual bandit techniques.

**Strengths:**

The paper proposes the AIO for the instruction optimization. The AIO provides some theoretical analysis and demonstrations.

**Weaknesses:**

1.	The evaluation benchmarks in this paper are not very common. The proposed improvements should be demonstrated in the general benchmarks, including the instructions for the following benchmarks.
2.	The proposed method is unlike other task-aware instruction optimization methods. As there are many methods and literatures in this field, it would be beneficial to compare the proposed methods with other baselines.
3.	Although the white box LLM and black box LLM show improvements, it is obvious that this combination will perform better than the white box LLM and black box LLM individually. So the more analyses on these component designs are needed.

**Questions:**

Please see the weakness section.

---

> ### Author Response · Authors · 2024-11-20
> **Thank you for the insightful review (1/3)**
>
> We sincerely thank the reviewer for your valuable questions and comments, which will certainly help improve the paper's presentation and make it more solid.
> We will provide a detailed response in the form of Q\&A. If you have any further questions or comments, please feel free to let us know. We would be happy to provide additional responses promptly. **Thank you!**
>
>
>
>
> ### Q1: The evaluation benchmarks in this paper are not common?
>
>
> *Please kindly refer to our response to Q2 for the reasons of our baseline selection.*
>
> Different from conventional instruction optimization settings, in this work, we address the challenge of automatically optimizing instructions for **a task-solving black-box LLM** in terms of the given **target task**. The process requires only a few exemplars as training data and eliminates the need for human expert intervention.
> This is an emerging topic of automatic instruction generation that has been explored in several related works (e.g., [2,9,13]). Unlike existing approaches, we propose utilizing a white-box LLM for instruction generation and optimization, coupled with LLM fine-tuning to effectively learn optimized instructions for highly complex modern black-box LLMs.
>
>
> As discussed in the Introduction and Related Works sections, the most relevant state-of-the-art works to this paper are [2,9], where a white-box LLM is also used as an instruction optimizer to tailor instructions specifically for the downstream task-solving black-box LLM.
> Therefore, for the datasets, we follow the most closely related works (e.g., [2,9,13]) by utilizing instruction induction tasks from [7], reasoning tasks from BigBench [1], and Chain-of-Thought (CoT) datasets [3,5,11] to benchmark our proposed AIO against baselines.
> *We would like to gently note that these datasets are both common and widely adopted in this line of research, particularly in works closely aligned with ours in terms of problem settings (e.g., [2,9,13]).*
>
>
>
> The instruction induction tasks from [7] and reasoning tasks from BigBench [1] are used to evaluate the zero-shot instruction induction (Subsec. 5.1) and few-shot instruction induction (Subsec. B.3) quality of the optimized instructions. In particular, since no auxiliary task-relevant information is provided to the task-solving black-box LLM, the zero-shot instruction induction results on these datasets [1,7] are widely adopted to assess instruction quality by closely related works (e.g., [2,4,8,9,13]).
> Additionally, the applied CoT datasets are also commonly used in the aforementioned related works (e.g., [2,9,13]), as they effectively test the instruction optimizers' ability to solve complex reasoning tasks, such as math problems.
> Therefore, our dataset selections are standard and widely adopted in closely related works, in order to demonstrate the effectiveness of AIO in terms of instructing the task-solving black-box LLM.
> We would also like to kindly mention that Reviewers oVqK and f3S6 have also expressed their acknowledgment of our comprehensive experiments.
>
>
> Meanwhile, we would like to kindly ask if the reviewer has any specific datasets in mind that should be included in the experiments.
> Due to computational costs (e.g., Claude daily token limit and GPU resources) and strict time constraints, if we are unable to provide the corresponding experimental results in a timely manner (e.g., if these datasets are challenging to adapt to our settings), we will ensure they are included in the revised version of our paper.
> Please also see our answer to Q2 in terms of our baseline selection.

---

> ### Author Response · Authors · 2024-11-20
> **Thank you for the insightful review (2/3)**
>
> ### Q2: It would be beneficial to compare the proposed methods with other task-aware instruction optimization baselines?
>
>
>
> As mentioned in our response to Q1, we primarily focus on the challenge of automatically optimizing instructions for a **task-solving black-box LLM** given the **target task**. This distinguishes our problem settings from those of conventional instruction optimization works.
> In this case, the most relevant works to ours are [2,9], which propose to deal with an analogous problem. From an in-context learning perspective instead, they apply a white-box LLM as an instruction generator to tailor instructions specifically for the downstream task-solving black-box LLM.
> Therefore, in our initial submission, *we have included these two works (InstructZero [2], INSTINCT [9]) as our baselines* to emphasize the advantages of LLM fine-tuning over their in-context learning approaches under our instruction optimization settings, given the superior representation and learning capabilities provided by LLM fine-tuning.
>
>
> Meanwhile, there is also a line of research (e.g., [12,13]) that employs a black-box LLM instead for instruction generation, resulting in a pipeline involving two black-box LLMs. In this setup, the instruction-generating black-box LLM perceives the feedback of the task-solver black-box LLM, and optimizes its instructing strategy from an in-context learning perspective.
> Since these methods are also commonly adopted by our closely related works [2,9], we have also included APE [13] and ProTeGi [12] as our baselines in our initial submission.
>
>
> *Therefore, we would like to gently note that our choice of baselines is standard in this line of research, including state-of-the-art baselines closely related to this paper, as well as common baselines adopted by other existing works in this research direction.*
>
>
>
>
> As suggested by the reviewer, we have included an additional baseline, EvoPrompt [6], which utilizes evolutionary algorithms to refine LLM-generated instructions in an in-context learning manner.
> Different from our original problem settings, where instructions are generated from scratch, EvoPrompt requires initial instructions. In this case, we follow the settings in the original paper [6] and the official source code of EvoPrompt, by using instructions generated by APE [13] as the initial instructions. We compare AIO with EvoPrompt, in terms of zero-shot instruction induction on four BigBench [1] tasks, as BigBench is also utilized in [6].
>
>
>
>
>
>
> | Method $\backslash$ Task | hyperbaton | navigate | movie recommendation | object counting
> |---  |--- |--- |--- |--- |
> | AIO       |    0.538    |   0.644  |   0.902  |   0.543
> | EvoPrompt |   0.526 |   0.627   | 0.895   |   0.466
>
>
> **Experiment results in comparison with EvoPrompt [6].**
>
>
>
>
> Based on the results in the above table, we see that our proposed AIO can manage to achieve generally better performance compared with EvoPrompt.
> Meanwhile, comparing with the two black-box LLM baselines in our initial submission: APE [13] and ProTeGi [12], EvoPrompt can achieve relatively better performance than them, by utilizing task prior knowledge provided by initial instructions as the starting point for optimization.
>
>
>
> Analogous to our answer in Q1, we would like to gently ask if the reviewer considers any specific baselines that should be included in the experiments.
> Given computational costs (e.g., Claude daily token limit and GPU resources) and strict time constraints, if we are unable to include the corresponding experimental results promptly (e.g., due to challenges in adapting these baselines and source code to our settings), we will certainly include them in the revised version of our paper.

---

> ### Author Response · Authors · 2024-11-20
> **Thank you for the insightful review (3/3)**
>
> ### Q3: Combination of the white box LLM and black box LLM? Analysis on component design?
>
>
>
> We would like to clarify that we are **not jointly using the white-box LLM and black-box LLM for instruction optimization**. In this paper, we tackle the challenge of automatically optimizing instructions for **a task-solving black-box LLM** with respect to the given **target task**.
> Therefore, *under our problem settings (Section 2), the black-box LLM is considered as part of the learning objective rather than the learning model, and we have no control over its parameters while can only interact with it through the API.*
> Therefore, based on our problem settings, which align with those of closely related works [2,9], the learning model will solely be the white-box LLM, while the black-box LLM will serve as part of the learning objective. Please kindly see our answer to Q1 for detailed discussions.
>
>
> In terms of the analysis of framework components, please kindly refer to our ablation study of AIO components, in Appendix Subsec. B.6, where we include two variants of AIO:
> (1) the first variant "AIO w/ MeZO" removes our decomposition of gradient flow (Remark 1), and directly approximates white-box LLM gradient with zeroth-order method MeZO [10]. (2) The second variant "AIO w/o TS Scheduling" abandons our TS scheduling module (Subsec. 4.1.2) and applies fully random perturbation vectors $\boldsymbol{z}\sim \mathcal{N}(0, \boldsymbol{I})$ for zeroth-order gradient approximation.
> This helps support our claim that the proposed gradient flow decomposition approach and the ZO black-box LLM gradient approximation method guided by TS are essential for AIO to achieve the optimal performance.
>
>
> Please also kindly refer to our answer to Q5 of Reviewer sk6i for additional sensitivity study on threshold parameter $\beta$ and number of chosen arms $B$ in each round.
>
>
>
> **REFERENCE**
>
> [1] BIG bench authors. Beyond the imitation game: Quantifying and extrapolating the capabilities of language models. Transactions on Machine Learning Research, 2023.
>
> [2] Lichang Chen, Jiuhai Chen, Tom Goldstein, Heng Huang, and Tianyi Zhou. Instructzero: Efficient instruction optimization for black-box large language models. In Forty-first International Conference on Machine Learning, 2024.
>
> [3] Karl Cobbe, Vineet Kosaraju, Mohammad Bavarian, Mark Chen, Heewoo Jun, Lukasz Kaiser, Matthias Plappert, Jerry Tworek, Jacob Hilton, Reiichiro Nakano, et al. Training verifiers to solve math word problems. arXiv preprint arXiv:2110.14168, 2021.
>
> [4] Chrisantha Fernando, Dylan Banarse, Henryk Michalewski, Simon Osindero, and Tim Rocktäschel. Promptbreeder: Self-referential self-improvement via prompt evolution. arXiv preprint arXiv:2309.16797, 2023.
>
> [5] Noa Garcia, Chentao Ye, Zihua Liu, Qingtao Hu, Mayu Otani, Chenhui Chu, Yuta Nakashima, and Teruko Mitamura. A dataset and baselines for visual question answering on art. In Computer Vision–ECCV 2020 Workshops: Glasgow, UK, August 23–28, 2020, Proceedings, Part II 16, pages 92–108. Springer, 2020.
>
> [6] Qingyan Guo, Rui Wang, Junliang Guo, Bei Li, Kaitao Song, Xu Tan, Guoqing Liu, Jiang Bian, and Yujiu Yang. Connecting large language models with evolutionary algorithms yields powerful prompt optimizers. In The Twelfth International Conference on Learning Representations, 2024.
>
> [7] Or Honovich, Uri Shaham, Samuel R Bowman, and Omer Levy. Instruction induction: From few examples to natural language task descriptions. arXiv preprint arXiv:2205.10782, 2022.
>
> [8] Wenyang Hu, Yao Shu, Zongmin Yu, Zhaoxuan Wu, Xiangqiang Lin, Zhongxiang Dai, See-Kiong Ng, and Bryan Kian Hsiang Low. Localized zeroth-order prompt optimization. arXiv preprint arXiv:2403.02993, 2024.
>
> [9] Xiaoqiang Lin, Zhaoxuan Wu, Zhongxiang Dai, Wenyang Hu, Yao Shu, See-Kiong Ng, Patrick Jaillet, and Bryan Kian Hsiang Low. Use your instinct: Instruction optimization for llms using neural bandits coupled with transformers. In Forty-first International Conference on Machine Learning, 2024.
>
> [10] Sadhika Malladi, Tianyu Gao, Eshaan Nichani, Alex Damian, Jason D Lee, Danqi Chen, and Sanjeev Arora. Fine-tuning language models with just forward passes. arXiv preprint arXiv:2305.17333, 2023.
>
> [11] Arkil Patel, Satwik Bhattamishra, and Navin Goyal. Are nlp models really able to solve simple math word problems? arXiv preprint arXiv:2103.07191, 2021.
>
> [12] Reid Pryzant, Dan Iter, Jerry Li, Yin Tat Lee, Chenguang Zhu, and Michael Zeng. Automatic prompt optimization with "gradient descent" and beam search. arXiv preprint arXiv:2305.03495, 2023.
>
> [13] Yongchao Zhou, Andrei Ioan Muresanu, Ziwen Han, Keiran Paster, Silviu Pitis, Harris Chan, and Jimmy Ba. Large language models are human-level prompt engineers. arXiv preprint arXiv:2211.01910, 2022.

---

> ### Author Response · Authors · 2024-11-27
> **Looking forward to your feedback**
>
> Dear Reviewer r8CJ,
>
> Thank you once again for your insightful review, comments, and questions, which have been crucial in improving the quality of our work.
> **As the PDF update will be disabled tomorrow, we kindly ask if our responses have adequately addressed your concerns regarding our work.**
> We would be sincerely grateful if you could review our responses and let us know if they sufficiently address your concerns. Any further comments or suggestions you may have would be greatly valued.
> **The current rebuttal discussions and additional empirical results have been incorporated into the manuscript**. We will continue to update the manuscript if you have any further comments or suggestions.
>
> For the response to your questions and concerns:
>
> 1. In Q1, we clarified **our choice of datasets** used in our experiments, where we follow common practices of closely related works in this line of research, in order to evaluate the quality of our optimized instructions.
>
>
>
> 2. In Q2, we clarified **our choice of baselines** used in our experiments by including closely related baselines with analogous problem settings to ours, which is a common practice in this line of research. Additionally, as requested by the reviewer, we provided additional empirical results by comparing our approach with **an additional baseline "EvoPrompt"**, which leverages "initial instructions" as task prior information for instruction optimization.
>
>
> 3. In Q3, we clarified that *we are not jointly using the white-box LLM and black-box LLM for instruction optimization in our problem settings*.
> Since we have no control over the black-box LLM parameters, interaction will be restricted to the black-box LLM API.
> Analogous to existing closely related works in this line of research, **the black-box LLM is treated as part of the learning objective rather than the learning model.**
>
>
>
>
>
>
> **We kindly invite you to refer to our updated manuscript for your reference.** Thank you once again for your time and dedication to the review process.
>
>
>
> Sincerely,
>
> Authors

---

> ### Author Response · Authors · 2024-12-02
> **Final day of the discussion phase: Please kindly check our response.**
>
> Dear Reviewer r8CJ,
>
> We would like to sincerely thank you again for your time and effort in reviewing our submission. Your feedback has been invaluable in improving the quality of our work, and we have carefully revised our manuscript as well as uploaded the latest version based on your comments.
>
> As tomorrow is the final day of the author-reviewer discussion phase, we kindly ask if you could review our responses to your comments and let us know if they address your concerns. We would greatly appreciate any additional feedback or suggestions you may have.
>
> Thank you again for your dedication to the review process, and we look forward to hearing from you.
>
> Best regards,
>
> Authors

---

### Official Review · Reviewer_oVqK · 2024-11-03

**Soundness:** 3
**Presentation:** 3
**Contribution:** 3
**Rating:** 6
**Confidence:** 3

**Summary:**

The paper proposes an instruction optimization framework called Automatic Instruction Optimizer (AIO) for black-box Large Language Models (LLMs). LLMs often rely on precise, task-relevant instructions to achieve high-quality outputs, but manual optimization is challenging, especially for black-box models where internal parameters are inaccessible. AIO addresses this by employing a white-box LLM, which can be fine-tuned to generate optimized instructions based on task-specific information and feedback from the black-box LLM. The optimization process involves a novel zeroth-order gradient approximation method, aided by Thompson Sampling, which bypasses the need for direct access to gradients, enabling effective fine-tuning. Extensive experiments demonstrate that AIO not only improves task performance but also reduces API token costs. Compared to existing methods, AIO achieves better performance on various tasks and maintains efficiency through the adaptive reuse of feedback. This framework represents a significant advancement in automating instruction generation, enhancing the usability and adaptability of black-box LLMs across diverse tasks​

**Strengths:**

+ The paper introduces a unique framework, Automatic Instruction Optimizer (AIO), that optimizes instructions for black-box LLMs without requiring access to internal parameters, which is a critical advantage in real-world applications where only API-based models are available.
+ AIO’s use of zeroth-order gradient approximation with Thompson Sampling is innovative, as it enables effective instruction fine-tuning without direct gradient access, thereby overcoming a major limitation in black-box model optimization.
+ The framework significantly reduces API token consumption by reusing black-box LLM feedback adaptively, making it a cost-effective solution compared to existing in-context learning or manual tuning methods.
+ The paper also explores lightweight PEFT variants (e.g., LoRA and Linear Probing) that maintain performance with fewer parameters, showcasing AIO’s flexibility for different fine-tuning needs.

**Weaknesses:**

+ The experiments mainly use llama+API pairing, which may not fully represent the framework’s effectiveness across a broader range of LLMs. I suggest the authors do more combinations of white-box LLMs, e.g., Mistral, for making the study of the method more comprehensive.

**Questions:**

The paper focuses on optimizing instructions based on task-specific feedback from a black-box LLM; how well does the optimized instruction generalize to related tasks or domains, especially in cases where task-specific data is sparse?

---

> ### Author Response · Authors · 2024-11-20
> **Thank you for the insightful review (1/2)**
>
> We sincerely thank the reviewer for your valuable questions and comments, which will certainly help improve the paper's presentation and make it more solid.
> We will provide a detailed response in the form of Q\&A. If you have any further questions or comments, please feel free to let us know. We would be happy to provide additional responses promptly. **Thank you!**
>
>
>
> ### Q1: More combinations of white-box LLMs, e.g., Mistral?
>
> We agree with the reviewer that including additional experiments with other types of white-box LLMs can benefit the audience. We conduct experiments with the LoRA module on two additional types of white-box LLMs: Mistral-7B-Instruct-v0.2 and Qwen2.5-7B-Instruct.
> With black-box LLM being Claude-3-Sonnet, we have zero-shot instruction induction results shown in the table below.
>
>
>
>
>
>
> | White-box LLM  $\backslash$ Task  | larger animal | navigate | sentiment | movie recommendation
> |---  |--- |--- |--- |--- |
> |  Llama-3-8B-Instruct   |   0.950    |   0.627  |   0.947  |   0.883
> |  Mistral-7B-Instruct-v0.2 |    0.890    |   0.634  |   0.907  |   0.874
> | Qwen2.5-7B-Instruct |    0.919    |   0.682 |   0.922  |   0.835
>
>
> **Experiment results with different kinds of white-box LLMs.**
>
>
>
>
> Based on the experimental results, we see that our proposed AIO framework achieves promising performance with other types of white-box LLMs other than the Llama family. Meanwhile, we also would like to mention that Llama 3 (Llama-3-8B-Instruct) generally retains a slight advantage over the other two white-box LLMs. One possible reason is that Llama 3 consists of 8 billion parameters, slightly more than the other two 7-billion-parameter models. This can provide an advantage in terms of the representation power to some extent.
>
>
>
>
>
>
>
>
>
>
>
>
> ### Q2: Generalize optimized instruction to related tasks or domains, especially when task-specific data is sparse?
>
>
>
>
> To begin with, we would like to recall that this work focuses on the problem of automatically optimizing instructions for a task-solving black-box LLM given a target task. The instruction generation and optimization require only a few exemplars as training data and do not involve human expert intervention.
> This problem is an emerging topic in the field of automatic instruction generation, with several related works (e.g., [2,7,8]). Different from existing works, in this paper, we propose leveraging a white-box LLM for instruction generation and optimization, combined with LLM fine-tuning to effectively learn optimized instructions for modern black-box LLMs of extreme complexity.
> However, while this paper focuses on optimizing instructions for a given target task, our AIO framework can be potentially extended to generalize across multiple related tasks or domains.
>
>
>
> **Part 1: Transfer learning and domain adaptation.**
>
> First, we can leverage ideas from transfer learning and domain adaptation to address this issue. Here, we can consider two related domains (tasks): the source domain and the target domain.
> The source domain will refer to the domain that we train the original instruction optimizer (white-box LLM), typically having larger amount of training data (exemplars) than the target domain.
> Afterwards, to adapt the trained white-box LLM to the target domain, we can apply few-shot domain adaptation techniques (e.g., [4]) to efficiently fine-tune the white-box LLM parameters for the target domain, even with a relatively smaller number of samples.
> In the meantime, in the context of transfer learning, we can possibly have additional theoretical insights. This is because the generalization loss on the target domain can be theoretically upper bounded by the empirical loss on the source domain, the learning power of the neural model, and the discrepancy between the source and target domains [1].

---

> ### Author Response · Authors · 2024-11-20
> **Thank you for the insightful review (2/2)**
>
> **Part 2: Meta-learning.**
>
> On the other hand, we can leverage ideas from meta-learning [5] to develop an instruction "meta-optimizer", which is capable of abstracting high-level information across multiple domains and generating high-quality meta-instructions (analogous to meta-prompts [6]).
> Using this approach, the "meta-optimizer" can be adapted to downstream tasks with only a few exemplars from the target task, without significant modifications of the AIO parameters.
> The performance of this approach is expected to depend on the neural model's learning capacity and the discrepancy among different tasks [3]. With a white-box LLM as the learning model with fine-tuning, providing sufficient representation power, AIO can help acquire high-level instruction knowledge using meta-learning techniques.
>
>
>
> Meanwhile, as mentioned in our experimental settings (Appendix Subsec. B.2), we use $20$ training exemplars, a reasonably small amount of training data, as the reference for the white-box LLM to generate and optimize instructions.
> Regarding the reviewer's question regarding data scarcity, we include additional zero-shot instruction induction experiments with varying exemplar quantities and the LoRA module. This is to investigate how the performance changes when altering the number of exemplars $|\mathcal{D}\_{\text{Train}}|$ for AIO, in terms of instruction generation and optimization.
>
>
>
>
>
>
>
> | Task $\backslash$ Number of exemplars | $5$ | $10$ | $20$ | $30$
> |---  |--- |--- |--- |--- |
> |  Larger animal  |    0.868    |  0.921  |   0.950 |   0.947
> |  Navigate |   0.622  |   0.634   |  0.627  |  0.681
>
> **Experiment results with different numbers of exemplars $|\mathcal{D}\_{\text{Train}}|$.**
>
>
>
>
> From the results above, we observe that the "larger animal" task can be more sensitive to the number of exemplars $|\mathcal{D}_{\text{Train}}|$. However, providing as few as $|\mathcal{D}\_{\text{Train}}| = 10$ query-label pairs as exemplars, which is a modest quantity, will allow AIO to achieve relatively promising performance.
> Meanwhile, we see that for the "navigate" task, a small number of $|\mathcal{D}\_{\text{Train}}| = 5$ exemplars can lead to promising results, which shows that it is more stable under the sparse data settings.
> Therefore, when fine-tuning AIO from scratch, the overall performance of AIO can possibly be influenced by data scarcity, with its impact varying based on the specific application scenarios of practitioners.
>
>
>
>
> We sincerely thank the reviewer for suggesting this intriguing idea. As similar to existing works (e.g., [2,7,8]), we focus on automatically optimizing instructions for the target task in this paper, we consider this topic a promising and interesting future direction.
> We will include above discussion to the manuscript after our discussion concludes.
>
>
> **REFERENCE**
>
> [1] Shai Ben-David, John Blitzer, Koby Crammer, Alex Kulesza, Fernando Pereira, and Jennifer Wortman Vaughan. A theory of learning from different domains. Machine learning, 79:151–175, 2010.
>
> [2] Lichang Chen, Jiuhai Chen, Tom Goldstein, Heng Huang, and Tianyi Zhou. Instructzero: Efficient instruction optimization for black-box large language models. In Forty-first International Conference on Machine Learning, 2024.
>
> [3] Qi Chen, Changjian Shui, and Mario Marchand. Generalization bounds for meta-learning: An information-theoretic analysis. Advances in Neural Information Processing Systems, 34:25878–25890, 2021.
>
> [4] Alexandra Chronopoulou, Matthew E Peters, and Jesse Dodge. Efficient hierarchical domain adaptation for pretrained language models. In Proceedings of the 2022 Conference of the North American Chapter of the Association for Computational Linguistics: Human Language Technologies, pages 1336–1351, 2022.
>
> [5] Chelsea Finn, Pieter Abbeel, and Sergey Levine. Model-agnostic meta-learning for fast adaptation of deep networks. In International conference on machine learning, pages 1126–1135. PMLR, 2017.
>
> [6] Yutai Hou, Hongyuan Dong, Xinghao Wang, Bohan Li, and Wanxiang Che. Metaprompting: Learning to learn better prompts. In Proceedings of the 29th International Conference on Computational Linguistics, pages 3251–3262, 2022.
>
> [7] Xiaoqiang Lin, Zhaoxuan Wu, Zhongxiang Dai, Wenyang Hu, Yao Shu, See-Kiong Ng, Patrick Jaillet, and Bryan Kian Hsiang Low. Use your instinct: Instruction optimization for llms using neural bandits coupled with transformers. In Forty-first International Conference on Machine Learning, 2024.
>
> [8] Yongchao Zhou, Andrei Ioan Muresanu, Ziwen Han, Keiran Paster, Silviu Pitis, Harris Chan, and Jimmy Ba. Large language models are human-level prompt engineers. arXiv preprint arXiv:2211.01910, 2022.

---

> ### Author Response · Authors · 2024-11-27
> **Looking forward to your feedback**
>
> Dear Reviewer oVqK,
>
> Thank you once again for your insightful review, comments, and questions, which have been crucial in improving the quality of our work.
> **As the PDF update will be disabled tomorrow, we kindly ask if our responses have adequately addressed your concerns regarding our work.**
> We would be sincerely grateful if you could review our responses and let us know if they sufficiently address your concerns. Any further comments or suggestions you may have would be greatly valued.
> **The current rebuttal discussions and additional empirical results have been incorporated into the manuscript**. We will continue to update the manuscript if you have any further comments or suggestions.
>
> For the response to your questions and concerns:
>
> 1. In Q1, we provided additional empirical results on **other families of white-box LLMs (Mistral and Qwen)**, as suggested by the reviewer.
>
> 2. In Q2, we discussed potential future directions and extensions of this work, to **generalize optimized instructions across related tasks or domains**. Additionally, we included discussions on data scarcity, and presented empirical results for varying numbers of exemplars available during the instruction optimization process.
>
>
>
>
> **We kindly invite you to refer to our updated manuscript for your reference.** Thank you once again for your time and dedication to the review process.
>
>
>
> Sincerely,
>
> Authors

---

> ### Author Response · Authors · 2024-12-02
> **Final day of the discussion phase: Please kindly check our response.**
>
> Dear Reviewer oVqK,
>
> We would like to sincerely thank you again for your time and effort in reviewing our submission. Your feedback has been invaluable in improving the quality of our work, and we have carefully revised our manuscript as well as uploaded the latest version based on your comments.
>
> As tomorrow is the final day of the author-reviewer discussion phase, we kindly ask if you could review our responses to your comments and let us know if they address your concerns. We would greatly appreciate any additional feedback or suggestions you may have.
>
> Thank you again for your dedication to the review process, and we look forward to hearing from you.
>
> Best regards,
>
> Authors

---

### Official Review · Reviewer_sk6i · 2024-11-08

**Soundness:** 2
**Presentation:** 2
**Contribution:** 2
**Rating:** 5
**Confidence:** 4

**Summary:**

This work proposed to fine-tune a white-box LLM to generate optimal instruction for the black-box LLM for downstream tasks. Under the assumption of linear optimization landscape of $\phi$, a zero-order gradient approximation approach was leveraged to update the parameters, in conjunction with a decision-making process that employed contextual bandits to select the most informative directions of gradient perturbation. Experiments of $15$ instruction induction and reasoning tasks showed the superior performance of the proposed algorithm.

**Strengths:**

1. The idea of automatic instruction optimizer is important to domain experts who do not have enough experience in prompting engineering. An added benefit would be increased transparency and trustworthiness between human and LLM agents.

2. The zero-order optimization framework with gradient approximation and decomposition is reasonable when involving many parameters.

**Weaknesses:**

1. The structure can be significantly improved for better clarification and illustration. There are many re-defined notations and formulations, such as $\phi$, $Theta_t$ and $\mathbf{z} \sim \mathcal{N}(0, I)$. Showing the trivial calculation process makes it difficult to grasp the core structure of the algorithm. It is suggested to first give the overall framework of the algorithm and then briefly present the calculation details.

2. The motivation of introducing Thompson sampling, in consideration of the zero-order gradient approximation, is confused. In addition, additional hyperparameters $\beta$ and $B$ were introduced without enough sensitivity analysis in experiments.

3. There is missing legend in Figure 3.

**Questions:**

1. Is the task context known by the white-box LLM? How can domain expert be involved in the design of instruction?

2. Does the Thompson sampling introduces bias of gradient estimation? What is the influence of hyperparameters $\beta$ and $B$?

---

> ### Author Response · Authors · 2024-11-20
> **Thank you for the insightful review (1/4)**
>
> We sincerely thank the reviewer for your valuable questions and comments, which will certainly help improve the paper's presentation and make it more solid.
> We will provide a detailed response in the form of Q\&A. If you have any further questions or comments, please feel free to let us know. We would be happy to provide additional responses promptly. **Thank you!**
>
>
>
>
>
>
> ### Q1: Structure can be improved for better presentation?
>
>
>
> We sincerely appreciate the reviewer's constructive and valuable suggestions in terms of improving the manuscript presentation.
> Based on your comments, we have updated the manuscript from the following perspectives: (1) We improve the presentation of our pipeline illustration, such as enlarging the fonts and adding additional references to specific paper contents; (2)
> Following the reviewer's suggestion, at the beginning of Section 4, we have included a paragraph briefly introducing the main framework pipeline and components, in order to help readers grasp the core structure of the proposed framework;
> (3) We have also improved the narrative of our calculation process to provide a clearer presentation, for a potentially broad audience that can be unfamiliar with zeroth-order gradient approximation techniques or Contextual Bandits.
> Additional detailed calculation steps will be elaborated later in the pseudo-code (Algorithm 1) instead.
> (4) The missing legend and some typos have also been corrected.
>
>
> *For your reference, please kindly refer to the updated manuscript that has been uploaded to the OpenReview platform.*
> If you have any additional comments on improving our paper presentation, please kindly let us know. We will try our best to address them in a timely manner. Thank you!
>
>
>
>
>
>
>
>
>
> ### Q2: Missing legend in Figure 3?
>
> We sincerely thank the reviewer for identifying this issue, and we have updated the manuscript accordingly.
> *For your reference, please kindly refer to the updated manuscript that has been uploaded to the OpenReview platform.*
>
>
>
>
>
>
>
>
>
> ### Q3: Is the task context known by the white-box LLM? How can domain expert be involved in the design of instruction?
>
> As shown in our pipeline illustration (the Generation Template), we only need a few exemplars from the target task as the reference to the white-box LLM for instruction generation and optimization. In this case, we do not need specific human-crafted task context for our AIO, which avoids the need for human expert intervention. This emerging topic in automatic instruction generation has been explored in several related works (e.g., [3,5,11]).
> Unlike prior approaches, we propose utilizing a white-box LLM for instruction generation and optimization, paired with LLM fine-tuning to efficiently learn optimized instructions for modern, highly complex black-box LLMs.
>
>
> On the other hand, if domain experts (e.g., human engineers) or domain knowledge (e.g., textual task descriptions or narratives) are available, there are several ways of integrating them to our instruction optimization process.
>
>
>
> **Part 1: Task narrative or description.**
>
> First, if task descriptions are provided to AIO as prior knowledge, we can incorporate this information into the instruction generation template (left-hand side of Figure~1). In this case, the input to the white-box LLM will include both the task exemplars and the textual task description. We conduct additional experiments in terms of zero-shot instruction induction on four BigBench tasks with the LoRA module, incorporating the one-sentence task descriptions provided by the BigBench dataset into our generation template.
>
>
>
>
>
>
>
> | Method $\backslash$ Task  | epistemic reasoning | logical fallacy | hyperbaton | movie recommendation
> |---  |--- |--- |--- |--- |
> |  AIO             |    0.719    |   0.836  |   0.527  |   0.883
> | AIO w/ Task Info |   0.811  |   0.872   | 0.639 |   0.895
>
>
>
> **Experiment results with additional textual task description information.**
>
>
>
>
> We see that involving additional textual task information into the instruction generation and optimization process can generally improve performance, particularly for logical reasoning tasks (e.g., epistemic reasoning, logical fallacy, hyperbaton), as task descriptions can help provide extra background information to prevent potential misinterpretations of exemplars.

---

> ### Author Response · Authors · 2024-11-20
> **Thank you for the insightful review (2/4)**
>
> **Part 2: Human experts.**
>
> When human experts are available, following the idea of "Human-in-the-loop" [9], we can involve these experts to help AIO in terms of instruction optimization and evaluation.
> For instance, in terms of instruction improvement for medical diagnostics, AIO can generate instructions such as "Please analyze patient symptoms to identify possible illnesses." Human evaluators could define metrics emphasizing clarity (e.g., suggesting tools like "blood test" or "MRI scan") or appropriateness (e.g., ensuring instructions do not encourage unsafe practices).
> This can enhance AIO's applicability in sensitive fields where clarity and accuracy are crucial. Compared to composing instructions from scratch that requires more efforts, this formulation can help reduce the workload of human experts.
>
>
>
> Meanwhile, human experts can be involved into the instruction evaluation process. In this case, AIO can be requested to generate multiple candidate instructions after fine-tuning, and then human experts can proceed to examine which of the candidates is most appropriate from multiple dimensions, such as interpretability. This will also help to enhance AIO's applicability for real-world scenarios.
>
>
> We would like to thank the reviewer for suggesting this intriguing idea. While this paper focuses on automatically optimizing instructions with only task exemplars, we consider these topics as promising and exciting directions for future exploration, and we will include this discussion to the manuscript after our discussion concludes.
>
>
>
>
>
> ### Q4: Motivation of introducing Thompson sampling? Does the Thompson sampling introduces bias of gradient estimation?
>
>
> **Part 1: Motivations.**
>
>
> As mentioned in our manuscript (paragraph below Remark 1), conventional zeroth-order (ZO) gradient approximation methods (e.g., ZO-SGD) have a potential drawback: the perturbation vectors $\boldsymbol{z}$ are randomly sampled. Consequently, their gradient approximation directions in the optimization landscape are random, which can result in inefficient gradient estimation process. To overcome this challenge, numerous ZO methods have been proposed to incorporate guided or chosen directions for gradient optimization, enabling more efficient estimation of the target gradient (e.g., [2,7]).
>
>
>
> In this work, we propose reusing collected feedback by framing the ZO-based fine-tuning process as an online sequential decision-making problem, and applying Contextual Bandit techniques to effectively identify beneficial perturbation directions worth exploring.
> With the assumed linear optimization landscape as in existing works (e.g., [6,8]), we apply linear Thompson Sampling to leverage previously collected information, which includes arms (previous gradient approximation directions) and corresponding rewards (benefits of going along these directions), to achieve a more efficient gradient estimation (Subsec. 4.1.2).
> Within the linear optimization landscape, our TS model aids in selecting gradient directions for white-box LLM fine-tuning, by properly reducing the possibility of choosing low-value directions (e.g., directions that are orthogonal to the true gradient, which can provide limited information for gradient approximation).
> This approach helps reduce the validation loss through instruction generation and ZO approximation direction selection, aligned with our formulation of the arm reward (Eq. 8).
>
>
>
>
>
> **Ablation study on AIO components.**
> We also would like to mention that the effectiveness of our TS modeling is supported by our ablation study (Appendix Subsec. B.6), where we compare our AIO framework with two alternatives: (1) "AIO w/ MeZO", where MeZO [6] is used directly for approximating white-box LLM parameter gradients instead of our gradient decomposition formulation (Remark 1). (2) "AIO w/o TS Scheduling", where Thompson Sampling is not applied for selecting perturbation directions, and completely random perturbation vectors $\boldsymbol{z} \sim \mathcal{N}(0, \boldsymbol{I})$ are used for zeroth-order gradient approximation as in Eq. 5.
> We observe that our proposed AIO with TS-aided ZO gradient approximation generally achieves better performance, compared with the two baselines with substituted modules.

---

> ### Author Response · Authors · 2024-11-20
> **Thank you for the insightful review (3/4)**
>
> **Part 2: Bias from TS.**
>
>
>
> *By balancing exploitation and exploration, TS will not introduce considerable bias.*
> Under the settings of gradient approximation, our proposed TS-based zeroth-order gradient approximation method itself does not inherently introduce considerable bias in terms of gradient estimation, as Thompson Sampling techniques can naturally tackle the exploitation-exploration dilemma [1,10] by exploring various gradient directions instead of greedily exploiting only certain ones. Here, TS model will choose from sampled candidate approximation directions, instead of actually altering the direction vector value.
> Meanwhile, in the short term, it is possible that TS may introduce temporary bias if it focuses on some perturbation directions that can lead to high rewards (i.e., directions that can help reduce validation loss, Eq. 8). Due to the *bias-variance trade-off* [4], the temporary bias can also be beneficial under our gradient approximation settings, and we will elaborate on this point in the next paragraph.
> On the other hand, with only a few gradient directions applied for LLM fine-tuning [6] in each optimization round, directly applying conventional unbiased estimators with randomly sampled directions can possibly result in high variance in terms of gradient estimation [2], which also reflects the *bias-variance trade-off* in terms of zeroth-order gradient approximation.
>
>
>
> *We introduce the TS to balance the "bias" and "variance" trade-off for ZO gradient approximation, under query-limited scenarios.*
> Here, we would like to mention that state-of-the-art ZO gradient approximation methods with a guided approximation process (e.g., [7]) can also introduce temporary bias in terms of gradient direction selection. However, they have been shown highly effective and efficient, particularly under sample-efficient settings, due to their ability of involving informative gradient approximation directions to balance "bias" and "variance" for gradient approximation.
> As a result, due to the bias-variance trade-off, it can lead to the case that *no algorithm can serve as a universal solution to various application scenarios of zeroth-order approximation.*
> Therefore, practitioners need to tailor solutions to their specific application scenarios.
> In our case, by balancing high-reward directions (exploitation) and sampling TS parameters from the posterior (exploration), our TS-aided approximation can help stabilize the gradient estimates, making it advantageous in query-limited scenarios like ours.
> In particular, by leveraging a highly efficient linear TS model, the arm selection process ($\sim 0.4$ seconds per round) will be fast, ensuring the efficiency in terms of gradient direction selection.
>
>
>
> **Ablation study on TS-aided zeroth-order gradient approximation.**
> The effectiveness of our modeling is also validated by our ablation study (Appendix Subsec. B.6), where we include a variant, "AIO w/o TS Scheduling," for comparison. Instead of using TS, "AIO w/o TS Scheduling" applies randomly chosen perturbation vectors $\boldsymbol{z} \sim \mathcal{N}(0, \boldsymbol{I})$ for zeroth-order gradient approximation, as described in Eq. 5. In contrast, our AIO with TS-aided ZO gradient approximation achieves better performance by strategically selecting informative gradient approximation directions, based on collected information and knowledge from the optimization landscape.
>
>
>
> Thank you for initiating this insightful discussion. We will certainly include the above points into the manuscript once our discussion concludes.

---

> ### Author Response · Authors · 2024-11-20
> **Thank you for the insightful review (4/4)**
>
> ### Q5: Parameter study for $\beta$ and $B$?
>
>
> We agree with the reviewer that it will be beneficial to include additional study for parameters $\beta$ and $B$ of AIO. Here, additional experiment results for zero-shot instruction induction, on the "Larger animal" and "Navigate" tasks with the LoRA module, are presented in the two tables below.
>
>
>
>
> | Task $\backslash$ $B$ value | $1$ | $2$ | $3$ | $4$
> |---  |--- |--- |--- |--- |
> |  Larger animal    |   0.915    |  0.931  |   0.950 |   0.941
> | Navigate |   0.610  |   0.632   | 0.627 |   0.664
>
> **Experiment results with different numbers of chosen arms ($B$ values) in each round.**
>
>
>
> | Task $\backslash$ $\beta$ value | 1 | 10 | 100 | $\infty$
> |---  |--- |--- |--- |--- |
> |  Larger animal    |    0.922    |   0.915 |   0.932 |   0.950
> |  Navigate         |   0.611  |   0.634   | 0.676 |  0.627
>
>
> **Experiment results with different threshold $\beta$ values.**
>
>
>
> For parameter $B$, we observe that setting $B=3$ can achieve promising performance, which can help balance computational (and token) costs with performance.
> For parameter $\beta$, it is recommended to start the tuning process with a large $\beta$, as suggested in our Appendix B.2, to effectively leverage past received records of the optimization landscape. On the other hand, a sufficiently small threshold $\beta$ will cause the method to degenerate into "AIO w/o TS Scheduling", as in our ablation study (Subsec. B.6). In this case, the TS model will be excluded from selecting ZO approximation directions, leading to relatively inferior performance.
>
>
>
>
> **REFERENCE**
>
>
> [1] Shipra Agrawal and Navin Goyal. Thompson sampling for contextual bandits with linear payoffs. In ICML, pages 127–135. PMLR, 2013.
>
> [2] HanQin Cai, Daniel McKenzie, Wotao Yin, and Zhenliang Zhang. Zeroth-order regularized optimization (zoro): Approximately sparse gradients and adaptive sampling. SIAM Journal on Optimization, 32(2):687–714, 2022.
>
> [3] Lichang Chen, Jiuhai Chen, Tom Goldstein, Heng Huang, and Tianyi Zhou. Instructzero: Efficient instruction optimization for black-box large language models. In Forty-first International Conference on Machine Learning, 2024.
>
> [4] Yifan Hu, Xin Chen, and Niao He. On the bias-variance-cost tradeoff of stochastic optimization. Advances in Neural Information Processing Systems, 34:22119–22131, 2021.
>
> [5] Xiaoqiang Lin, Zhaoxuan Wu, Zhongxiang Dai, Wenyang Hu, Yao Shu, See-Kiong Ng, Patrick Jaillet, and Bryan Kian Hsiang Low. Use your instinct: Instruction optimization for llms using neural bandits coupled with transformers. In Forty-first International Conference on Machine Learning, 2024.
>
> [6] Sadhika Malladi, Tianyu Gao, Eshaan Nichani, Alex Damian, Jason D Lee, Danqi Chen, and Sanjeev Arora. Fine-tuning language models with just forward passes. arXiv preprint arXiv:2305.17333, 2023.
>
> [7] Ruizhong Qiu and Hanghang Tong. Gradient compressed sensing: A query-efficient gradient estimator for high-dimensional zeroth-order optimization. In Forty-first International Conference on Machine Learning.
>
> [8] James C Spall. Multivariate stochastic approximation using a simultaneous perturbation gradient approximation. IEEE transactions on automatic control, 37(3):332–341, 1992.
>
> [9] Xingjiao Wu, Luwei Xiao, Yixuan Sun, Junhang Zhang, Tianlong Ma, and Liang He. A survey of human-in-the-loop for machine learning. Future Generation Computer Systems, 135:364–381, 2022.
>
> [10] Weitong Zhang, Dongruo Zhou, Lihong Li, and Quanquan Gu. Neural thompson sampling. In International Conference on Learning Representations, 2021.
>
> [11] Yongchao Zhou, Andrei Ioan Muresanu, Ziwen Han, Keiran Paster, Silviu Pitis, Harris Chan, and Jimmy Ba. Large language models are human-level prompt engineers. arXiv preprint arXiv:2211.01910, 2022.

---

> ### Author Response · Authors · 2024-11-27
> **Looking forward to your feedback**
>
> Dear Reviewer sk6i,
>
> Thank you once again for your insightful review, comments, and questions, which have been crucial in improving the quality of our work.
> **As the PDF update will be disabled tomorrow, we would like to gently ask if our responses have adequately addressed your concerns regarding our work.**
> We would be sincerely grateful if you could review our responses and let us know if they sufficiently address your concerns. Any further comments or suggestions you may have would be greatly valued.
> **The current rebuttal discussions and additional empirical results have been incorporated into the manuscript**. We will continue to update the manuscript if you have any further comments or suggestions.
>
> For the response to your questions and concerns:
>
> 1. In Q1, we summarized the **improvements and modifications** made to the updated manuscript, according to the reviewer's comments on our paper presentation.
>
> 2. In Q2, we addressed the issue of the missing legend in our initial manuscript.
>
> 3. In Q3, we clarified the **"task information"** used for instruction generation and optimization, mentioning that only task exemplars are utilized for our instruction optimization.
> On the other hand, we also included additional empirical results demonstrating that AIO performance can be further enhanced, by incorporating extra **task descriptions** as prior knowledge into the optimization process. In addition, we discussed potential future directions for this work, by including the incorporation of **human experts** into the optimization process.
>
>
> 4. In Q4, we clarified the **motivations and intuitions** behind our TS-aided ZO gradient approximation approach, and included discussions on the bias of our TS direction selection strategy.
>
> 5. In Q5, we conducted additional **parameter studies** on the threshold $\beta$ and the number of chosen arms $B$, as suggested by the reviewer.
>
>
>
>
> **We kindly invite you to refer to our updated manuscript for your reference.** Thank you once again for your time and dedication to the review process.
>
>
>
> Sincerely,
>
> Authors

---

> ### Author Response · Authors · 2024-12-02
> **Final day of the discussion phase: Please kindly check our response.**
>
> Dear Reviewer sk6i,
>
> We would like to sincerely thank you again for your time and effort in reviewing our submission. Your feedback has been invaluable in improving the quality of our work, and we have carefully revised our manuscript as well as uploaded the latest version based on your comments.
>
>
> As tomorrow is the final day of the author-reviewer discussion phase, we kindly ask if you could review our responses to your comments and let us know if they address your concerns. We would greatly appreciate any additional feedback or suggestions you may have.
>
> Thank you again for your dedication to the review process, and we look forward to hearing from you.
>
> Best regards,
> Authors

---

### Author Response · Authors · 2024-11-25
**Looking Forward to Your Valuable Feedback on Rebuttal and Manuscript Updates**

Dear Reviewers,

We sincerely thank you for your time and dedication to the review process. Your insightful feedback, comments, and questions have been invaluable in improving the quality of our work.
As the discussion period will **conclude in two days**, we would greatly appreciate it if you could review our responses and let us know whether they sufficiently address your concerns. Any further comments or suggestions you may have would be deeply appreciated.


**Changes to the Updated Manuscript and Summary of Our Added Experiments:**


1. Improvements made to the manuscript based on reviewers' comments:

- (1) *[Figure 1]* **Enhanced pipeline illustration**: Enlarged fonts, added references to specific sections, etc.

- (2) *[Beginning of Section 4]* **Added introductory paragraph at the beginning of Section 4**: Briefly introduces the main framework pipeline and components to help readers understand the core structure of the proposed framework.

- (3) *[Subsection 4.1]* **Improved narrative of the calculation process**: Provides clearer presentation for a broader audience, especially those unfamiliar with zeroth-order gradient approximation techniques or Contextual Bandits. Detailed calculation steps are included later in the pseudo-code (Algorithm 1).

- (4) *[Equation 1 and Figure 3]* **Corrected missing legend and typographical errors.**

- (5) *[Appendix B.7 - B.11]* **Incorporated experimental results from the rebuttal phase.**

- (6) **Included discussions from the rebuttal phase**:

  1. *[Appendix E]* **Future Directions**:
     - Task information for instruction generation and optimization, including involving human experts. [Reviewer sk6i]
     - Generalizing optimized instructions to related tasks or domains. [Reviewer oVqK]

  2. *[Appendix D]* **Discussions and Clarifications**:
     - Reasoning behind our TS-aided ZO approach, and discussions on the bias in TS direction selection strategy. [Reviewer sk6i]
     - Clarifications on our choice of datasets and baselines, which are similar to closely related works. [Reviewer r8CJ]
     - Clarification on model usage: The black-box LLM is treated as part of the learning objective rather than the learning model, consistent with closely related works. [Reviewer r8CJ]


2. Additional experiments (already included in the updated manuscript):

- [Reviewer sk6i] Enhanced AIO performance by incorporating **task descriptions** as prior knowledge into the optimization process.

- [Reviewer sk6i] **Parameter study** on the threshold $\beta$ and the number of chosen arms $B$.

- [Reviewer oVqK] Additional empirical results on **other families of white-box LLMs** (Mistral and Qwen).

- [Reviewer r8CJ] Comparison with an **additional baseline "EvoPrompt"**, which leverages "initial instructions" as task prior information for instruction optimization.

- [Reviewer f3S6] Experimental results on **transferring** fine-tuned instruction optimizers across **different black-box LLMs** (Claude 3.5 Sonnet, GPT-3.5 Turbo, GPT-4o).



**The current rebuttal discussions and additional empirical results have been incorporated into the manuscript. We kindly invite you to review our updated manuscript for your reference.**




Sincerely,

Authors

---

### Comment · Area_Chair_fJex · 2024-11-25
**Please actively participate in the reviewer-author discussion**

Dear Reviewers,

Thank you for your efforts and contribution to ICLR! The authors have posted their responses to your original comments. Only less than two days are left for the reviewer-author discussion. Given the current borderline ratings, your help and prompt responses are important. Please actively check the authors' responses and actively participate in the discussion.

Thanks!

Best regards,

Your AC

---

> ### Author Response · Authors · 2024-11-25
> **Thank you for the reminder!**
>
> Dear Area Chair,
>
> Thank you so much for your timely reminder and for your dedication to the review process!
>
> Best regards,
> Authors

---

### Meta-Review · Area_Chair_fJex · 2024-12-23

**Metareview:**

## Summary:
The paper introduces an Automatic Instruction Optimizer (AIO) framework designed for optimizing task-relevant instructions for black-box Large Language Models (LLMs), where internal parameters and gradients are inaccessible. AIO finetunes a white-box LLM (via parameter efficient fine-tuning) using Contextual Bandit (TS) based zeroth-order gradient approximation to adaptively adjust instructing strategies, based on feedback from the black-box LLM. Through experiments on instruction induction and reasoning tasks. AIO showcases effectiveness in automating instruction generation, enhancing the usability and adaptability of black-box LLMs across various tasks, leading to improved task performance and reduced API token costs. AIO follows the existing framework of instruction optimization but adopts a different zeroth order optimization, which offers a promising solution for optimizing task-aware instructions in the absence of detailed task-solver LLM information.

## Strengths:
1. AIO's use of zeroth-order gradient approximation with Thompson Sampling is innovative, enabling effective instruction fine-tuning without direct gradient access.
1. The paper explores lightweight PEFT variants, showcasing AIO's flexibility for different fine-tuning needs and maintaining performance with fewer parameters.
1. Comprehensive experiments validate the effectiveness of the proposed method, including diverse white-box and black-box LLM combinations, instruction optimization trajectory results, and comparisons to existing approaches like CoT.

## Weaknesses:
1. This paper's contribution is incremental to some extent: it follows the existing framework of white-box LLM + API (black-box) LLM framework that optimizes a few parameters of the white-box LLM  zeroth order optimization. The main difference is in the zeroth-order optimization approach, which is Thompson sampling (parametric prior and posterior) in this paper but Bayesian optimization (nonparametric prior and posterior) in previous works such as InstructZero. However, theoretical analysis and more motivations are needed to justify the TS approach is better than previous strategies.
1. The proposed approach mainly focuses on optimizing instructions for a specified task, while a novel extension is to generalize the instruction optimization for other tasks. While the authors proposed valuable ideas in the discussion, extensive study and experiments of them can greatly strengthen the novelty and contribution of this paper.
1. Different combinations of white-box LLM + black-box LLM need to be examined. In addition, fixing the white-box LLM but changing the black-box LLM can reveal the generalizability/transferability of the instructions while fixing the black-box LLM but changing the white-box LLM shed insights on which white-box LLMs are better instruction generators. These results should be compared with the zero/few-shot performance on the individual white-box LLM or the black-box LLM only.
1. The original paper lacks several ablation studies on the hyperparameters of TS, the number of exemplars, whether using the task information, etc. The authors' responses provide these results in some experiments. The authors are encouraged to provide complete results for these experiments in the next version.
1. Reviewers also raised concerns regarding the notations, equations, and figures, whose presentations can be improved.

## Decision:
The authors provided detailed clarifications and additional experimental results in the rebuttal, as requested by the reviewers. Reviewers did not participate in the author-reviewer discussion but one reviewer participated in the AC-reviewer discussion. Given the borderline ratings and the mixed comments, the meta-reviewer carefully read the paper and all the authors' responses. The authors put a great amount of effort into addressing the reviewers' concerns, including the additional experiments and revising the narratives. They also provided a nice summary of the updates for review convenience. However, the incremental contribution of the problem setup and the instruction optimization framework is still one of the key factors considered in the decision-making. Reviewers expect to obtain more novel insights in addition to empirical improvements in the previously studied datasets. Moreover, the rebuttal and discussion indicate that a great amount of additional improvement needs to be made to the current draft. The meta-reviewer believes another round of revision is necessary for this work. The authors are encouraged to improve the draft as suggested by the reviewers and meta-reviewer.

**Additional Comments On Reviewer Discussion:**

The authors provided detailed clarifications and additional experimental results in the rebuttal, as requested by the reviewers. Reviewers did not participate in the author-reviewer discussion but one reviewer participated in the AC-reviewer discussion. Given the borderline ratings and the mixed comments, the meta-reviewer carefully read the paper and all the authors' responses. The authors put a great amount of effort into addressing the reviewers' concerns, including the additional experiments and revising the narratives. They also provided a nice summary of the updates for review convenience. However, the incremental contribution of the problem setup and the instruction optimization framework is still one of the key factors considered in the decision-making. Reviewers expect to obtain more novel insights in addition to empirical improvements in the previously studied datasets. Moreover, the rebuttal and discussion indicate that a great amount of additional improvement needs to be made to the current draft. The meta-reviewer believes another round of revision is necessary for this work. The authors are encouraged to improve the draft as suggested by the reviewers and meta-reviewer.

---

### Decision · Program_Chairs · 2025-01-22

Reject